# Bayesian Pose Graph Optimization via Bingham Distributions and Tempered Geodesic MCMC

**Tolga Birdal[1,2]**      **Umut Şimşekli[3]**      **M. Onur Eken[1,2]**      **Slobodan Ilic[1,2]**

[1] CAMP Chair, Technische Universität München, 85748, München, Germany

[2] Siemens AG, 81739, München, Germany

[3] LTCI, Télécom ParisTech, Université Paris-Saclay, 75013, Paris, France

## Abstract

We introduce Tempered Geodesic Markov Chain Monte Carlo (TG-MCMC) algorithm for initializing pose graph optimization problems, arising in various scenarios such as SFM (structure from motion) or SLAM (simultaneous localization and mapping). TG-MCMC is first of its kind as it unites global non-convex optimization on the spherical manifold of quaternions with posterior sampling, in order to provide both reliable initial poses and uncertainty estimates that are informative about the quality of solutions. We devise theoretical convergence guarantees and extensively evaluate our method on synthetic and real benchmarks. Besides its elegance in formulation and theory, we show that our method is robust to missing data, noise and the estimated uncertainties capture intuitive properties of the data.

## 1   Introduction

The ability to navigate autonomously is now a key technology in self driving cars, unmanned aerial vehicles (UAV), robot guidance, augmented reality, 3D digitization, sensory network localization and more. This ubiquitous appliance is due to the fact that vision sensors can provide cues to directly solve 6DoF pose estimation problem and do not necessitate external tracking input, such as imprecise GPS, to ego-localize. Many of the problems in these domains can now be addressed by tailor-made pipelines such as SLAM (Simultaneous Localization and Mapping), SfM (Structure From Motion) or multi robot localization (MRL) [1, 2]. Nowadays, thanks to the resulting reliable estimates of rotations and translations, many of these pipelines exploit some form of an optimization, such as bundle adjustment (BA) [3] or 3D global registration [4, 5], that can globally consider the acquired measurements [6]. Holistically, these methods belong to the family of *pose graph optimization* (PGO) [7]. Unfortunately, many of PGO post-processing stages, which take in to account both camera poses and 3D structure, are too costly for online or even soft-realtime operation. This bottleneck demands good solutions for PGO initialization, that can relieve the burden of the joint optimization.

In this paper, we address the particular problem of initializing PGO, in which multiple local measurements are fused into a globally consistent estimate, without resorting to the costly bundle adjustment or optimization that uses structure. In specifics, let us consider a finite simple directed graph $G = (V, E)$, where vertices correspond to reference frames and edges to the available relative measurements as shown in Figures 1(a), 1(b). Both vertices and edges are labeled with rigid motions representing absolute and relative poses, respectively. Each absolute pose is described by a homogeneous transformation matrix $\{\mathbf{M}_i \in SE(3)\}_{i=1}^n$. Similarly, each relative orientation is expressed as the transformation between frames $i$ and $j$, $\mathbf{M}_{ij}$, where $(i, j) \in E \subset [n] \times [n]$. The labeling of the edges is such that if $(i, j) \in E$, then $(j, i) \in E$ and $\mathbf{M}_{ij} = \mathbf{M}_{ji}^{-1}$. Hence, we consider $G$ to be undirected. With a convention as shown in Figure 1(c), the link between absolute and relative transformations is encoded by the *compatibility constraint*:

$$\mathbf{M}_{ij} \approx \mathbf{M}_j \mathbf{M}_i^{-1}, \, \forall i \neq j \tag{1}$$

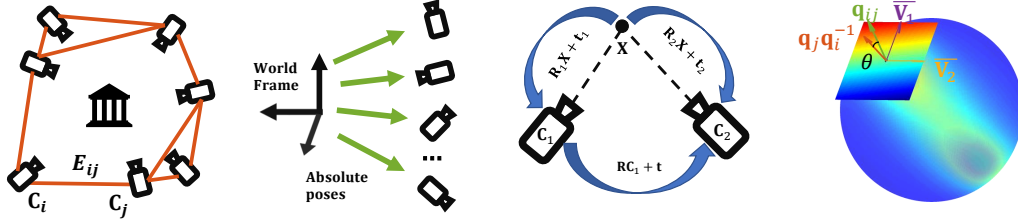

Figure 1: From left to right: **(a)** Initial pose graph of relative poses. **(b)** Absolute poses w.r.t. common reference frame. **(c)** Convention used to describe the pairwise relationships. **(d)** A sample Bingham distribution and the rotational components.

Primarily motivated by Govindu et. al. [8], *rigid-motion synchronization* initializes PGO by computing an estimate of the vertex labels $\mathbf{M}_i$ (absolute poses) given enough measurements of the ratios $\mathbf{M}_{ij}$. In other words, it tries to find the absolute poses that best fit the relative pairwise measurements. Typically, in order to remove the gauge freedom, one of the poses is set to identity $\mathbf{M}_0 = \mathbf{I}$ and the problem reduces to recovering $n-1$ absolute poses. The solution is the state of the art method to initialize, say SfM [1, 9, 10] thanks to the good quality of the estimates.

The PGO problem is often formed as non-convex optimization problems, opening up room for different formulations and approaches. Direct methods try to compute a good initial solution [11, 9, 12, 13], which are then refined by iterative techniques [14, 15]. Robustness to outlier relative pose estimates is also crucial for a better solution [16, 17, 10, 18, 2]. The structure of our peculiar problem allows for global optimization, when isotropic noise is assumed and under reasonable noise levels as well as well connected graph structures [11, 19, 20, 21, 22, 23]. It is also noteworthy that, even though the problem has been previously handled with statistical approaches [24], up until now, to the best of our knowledge, estimation of uncertainties in PGO initialization are never truly considered.

In this paper, we look at the graph optimization problem from a probabilistic point of view. We begin by representing the problem on the Cartesian product of the true Riemannian manifold of quaternions and Euclidean manifold of translations. We model rotations with Bingham distributions [25] and translation with Gaussians. The probabilistic framework provides two important features: (i) we can align the modes of the data (relative motions) with the posterior parameters, (ii) we can quantify the uncertainty of our estimates by using the posterior predictive distributions. In order to achieve these goals, we come up with efficient algorithms both for maximum a-posteriori (MAP) estimation and posterior sampling: 'tempered' geodesic Markov Chain Monte Carlo (TG-MCMC). Controlled by a single parameter, TG-MCMC can either work as a standard MCMC algorithm that can generate samples from a Bayesian posterior, whose entropy, or covariance, as well as the samples themselves, provide necessary cues for uncertainty estimation - both on camera poses and possibly on the 3D structure, or it can work as an optimization algorithm that is able to generate samples around *the global optimum* of the MAP estimation problem. In this perspective, TG-MCMC bridges the gap between geodesic MCMC (gMCMC) [26] and non-convex optimization, as we will theoretically present. In a nutshell, our contributions are as follows:

- Novel probabilistic model using Bingham distributions in pose averaging for the first time,
- Tempered gMCMC: Novel tempered Hamiltonian Monte Carlo (HMC) [27, 28, 29] algorithm for global optimization and sampling on the manifolds using the known geodesic flow,
- Theoretical understanding and convergence guarantees for the devised algorithm,
- Strong experimental results justifying the validity of the approach.

## 2 Preliminaries and Technical Background

**Notation and definitions:** $x \in \mathbb{R}$ is a scalar. We denote vectors by lower case bold letters $\mathbf{x} = (x_1 \cdots x_N) \in \mathbb{R}^N$. A square matrix $\mathbf{X} = (X_{ij}) \in \mathbb{R}^{N \times N}$. $\mathbf{I}_{N \times N}$ is the identity matrix. Rotations belong to the special orthogonal group $\mathbf{R} \in SO(3)$. With translations $\mathbf{t} \in \mathbb{R}^3$, they form the 3D special Euclidean group $SE(3)$. We also define an $m$-dimensional *Riemannian manifold* $\mathcal{M}$, endowed with a *Riemannian metric* $\mathbf{G}$ to be a smooth curved space, equipped with the inner product $\langle \mathbf{u}, \mathbf{v} \rangle_x = \mathbf{u}^T \mathbf{G} \mathbf{v}$ in the tangent space $\mathcal{T}_x \mathcal{M}$, embedded in an ambient higher-dimensional Euclidean

space $\mathbb{R}^n$. One such manifold is the unit hypersphere in $\mathbb{R}^d$: $\mathbb{S}^{d-1} = \{\mathbf{x} \in \mathbb{R}^d : \|\mathbf{x}\| = 1\} \subset R^d$. A vector $\mathbf{v}$ is said to be *tangent* to a point $\mathbf{x} \in \mathcal{M}$ if $\mathbf{x}^T \mathbf{v} = 0$. A tangent space is the set $\mathcal{T}_x$ of all such vectors: $\mathcal{T}_x = \{\mathbf{v} \in \mathbb{R}^d : \mathbf{x}^T \mathbf{v} = 0\}$. We define the geodesic on the manifold to be a constant speed, length minimizing curve between $\mathbf{x}, \mathbf{y} \in \mathcal{M}$, $\gamma : [0, 1] \to \mathcal{M}$, with $\gamma(0) = \mathbf{x}$ and $\gamma(1) = \mathbf{y}$.

**Quaternions:** A *quaternion* $\mathbf{q}$ is an element of Hamilton algebra $\mathbb{H}$, extending the complex numbers with three imaginary units $\mathbf{i}, \mathbf{j}, \mathbf{k}$ in the form $\mathbf{q} = q_1 \mathbf{1} + q_2 \mathbf{i} + q_3 \mathbf{j} + q_4 \mathbf{k} = (q_1, q_2, q_3, q_4)^T$, with $(q_1, q_2, q_3, q_4)^T \in \mathbb{R}^4$ and $\mathbf{i}^2 = \mathbf{j}^2 = \mathbf{k}^2 = \mathbf{ijk} = -\mathbf{1}$. We also write $\mathbf{q} := [a, \mathbf{v}]$ with the scalar part $a = q_1 \in \mathbb{R}$ and the vector part $\mathbf{v} = (q_2, q_3, q_4)^T \in \mathbb{R}^3$. The conjugate $\bar{\mathbf{q}}$ of the quaternion $\mathbf{q}$ is given by $\bar{\mathbf{q}} := q_1 - q_2 \mathbf{i} - q_3 \mathbf{j} - q_4 \mathbf{k}$. A versor or *unit quaternion* $\mathbf{q} \in \mathbb{H}_1$ with $1 \stackrel{!}{=} \|\mathbf{q}\| := \mathbf{q} \cdot \bar{\mathbf{q}}$ and $\mathbf{q}^{-1} = \bar{\mathbf{q}}$, gives a compact and numerically stable parametrization to represent orientation of objects in $\mathbb{S}^3$, avoiding gimbal lock and singularities [30, 31]. Identifying antipodal points $\mathbf{q}$ and $-\mathbf{q}$ with the same element, the unit quaternions form a double covering group of $SO(3)$. The non-commutative multiplication of two quaternions $\mathbf{p} := [p_1, \mathbf{v}_p]$ and $\mathbf{r} := [r_1, \mathbf{v}_r]$ is defined to be $\mathbf{p} \otimes \mathbf{r} = [p_1 r_1 - \mathbf{v}_p \cdot \mathbf{v}_r, p_1 \mathbf{v}_r + r_1 \mathbf{v}_p + \mathbf{v}_p \times \mathbf{v}_r]$. For simplicity we use $\mathbf{p} \otimes \mathbf{r} := \mathbf{p} \cdot \mathbf{r} := \mathbf{pr}$.

**Manifold of quaternions:** Unit quaternions form a hyperspherical manifold, $\mathbb{S}^3$, that is an embedded Riemannian submanifold of $\mathbb{R}^4$. This forms a Hausdorff space, where each point has an open neighborhood homeomorphic to the open N-dimensional disc, called an N-manifold. Due to the topology of the sphere, there is no unique way find a globally covering coordinate patch. It is hence common to use local exponential and logarithmic maps that can be sphere-specifically defined as: $\text{Exp}(\mathbf{x}, \mathbf{v}) = \mathbf{x} \cos(\theta) + \mathbf{v} \sin(\theta)/\theta$, where $\mathbf{v}$ denotes a tangent vector to $\mathbf{x}$. This property decorates quaternions with a known analytic geodesic flow, given by [26]:

$$\begin{bmatrix} \mathbf{x}(t) & \mathbf{v}(t) \end{bmatrix} = \begin{bmatrix} \mathbf{x}(0) & \mathbf{v}(0) \end{bmatrix} \begin{bmatrix} 1 & 0 \\ 0 & 1/\alpha \end{bmatrix} \begin{bmatrix} \cos(\alpha t) & -\sin(\alpha t) \\ \sin(\alpha t) & \cos(\alpha t) \end{bmatrix} \begin{bmatrix} 1 & 0 \\ 0 & \alpha \end{bmatrix} \tag{2}$$

where $\alpha \triangleq \|\mathbf{v}(0)\|$. It is also useful to think about a quaternion as the normal vector to itself, due to the unitness of the hypersphere. By this property, projection onto $\mathcal{T}_x$ reads $P(\mathbf{x}) = \mathbf{I} - \mathbf{xx}^T$ [26].

**The Bingham Distribution:** Derived from a zero-mean Gaussian, the Bingham distribution [25] is an antipodally symmetric probability distribution conditioned to lie on $\mathbb{S}^{d-1}$ with probability density function (PDF) $\mathcal{B} : \mathbb{S}^{d-1} \to R$:

$$\mathcal{B}(\mathbf{x}; \boldsymbol{\Lambda}, \mathbf{V}) = (1/F) \exp(\mathbf{x}^T \mathbf{V} \boldsymbol{\Lambda} \mathbf{V}^T \mathbf{x}) = (1/F) \exp\left(\sum_{i=1}^d \lambda_i (\mathbf{v}_i^T \mathbf{x})^2\right) \tag{3}$$

where $\mathbf{V} \in \mathbb{R}^{d \times d}$ is an orthogonal matrix ($\mathbf{VV}^T = \mathbf{V}^T \mathbf{V} = \mathbf{I}_{d \times d}$) describing the orientation, $\boldsymbol{\Lambda} = \text{diag}(0, \lambda_1, \cdots, \lambda_{d-1}) \in R^{d \times d}$ with $0 \geq \lambda_1 \geq \cdots \geq \lambda_{d-1}$ is the concentration matrix, and $F$ is a normalization constant. With this formulation, the mode of the distribution is obtained as the first column of $\mathbf{V}$. The antipodal symmetry of the PDF makes it amenable to explain the topology of quaternions, i. e., $\mathcal{B}(\mathbf{x}; \cdot) = \mathcal{B}(-\mathbf{x}; \cdot)$ holds for all $\mathbf{x} \in \mathbb{S}^{d-1}$. When $d = 4$ and $\lambda_1 = \lambda_2 = \lambda_3$, it is safe to write $\boldsymbol{\Lambda} = \text{diag}([1, 0, 0, 0])$. In this case, the logarithm of the Bingham density reduces to the dot product of two quaternions $\mathbf{q}_1 \triangleq \mathbf{x}$ and the mode of the distribution, say $\bar{\mathbf{q}}_2$. For rotations, this induces a metric, $d_{\text{bingham}} = (\mathbf{q}_1 \cdot \bar{\mathbf{q}}_2)^2 = \cos(\theta/2)^2$, that is closely related to the true Riemannian distance $d_{\text{riemann}} = \|\log(\mathbf{R}_1 \mathbf{R}_2^T)\| \triangleq 2\arccos(|\mathbf{q}_1 \bar{\mathbf{q}}_2|) \triangleq 2\arccos(\sqrt{d_{\text{bingham}}})$. Bingham distributions have been extensively used to represent distributions on quaternions [32, 33, 34]; however, to the best of our knowledge, never for the problem at hand.

## 3 The Proposed Model

We now describe our proposed model for PGO initialization. We consider the situation where we observe a set of noisy pairwise poses $\mathbf{M}_{ij}$, represented by *augmented quaternions* as $\{\mathbf{q}_{ij} \in \mathbb{S}^3 \subset \mathbb{R}^4, \mathbf{t}_{ij} \in \mathbb{R}^3\}$. The indices $(i, j) \in E$ run over the edges the graph. We assume that the observations $\{\mathbf{q}_{ij}, \mathbf{t}_{ij}\}_{(i,j) \in E}$ are generated by a probabilistic model that has the following hierarchical structure:

$$\mathbf{q}_i \sim p(\mathbf{q}_i), \qquad \mathbf{t}_i \sim p(\mathbf{t}_i), \qquad \mathbf{q}_{ij}| \cdot \sim p(\mathbf{q}_{ij}|\mathbf{q}_i, \mathbf{q}_j), \qquad \mathbf{t}_{ij}| \cdot \sim p(\mathbf{t}_{ij}|\mathbf{q}_i, \mathbf{q}_j, \mathbf{t}_i, \mathbf{t}_j), \tag{4}$$

where the *latent variables* $\{\mathbf{q}_i \in \mathbb{S}^3\}_{i=1}^n$ and $\{\mathbf{t}_i \in \mathbb{R}^3\}_{i=1}^n$ denote the true values of the *absolute poses* and *absolute translations* with respect to a common origin, corresponding to $\mathbf{M}_i$ of Eq. 1. Here, $p(\mathbf{q}_i)$ and $p(\mathbf{t}_i)$ denote the *prior distributions* of the latent variables, and the product of the densities $p(\mathbf{q}_{ij}|\cdot)$ and $p(\mathbf{t}_{ij}|\cdot)$ form the *likelihood* function.

By respecting the natural manifolds of the latent variables, we choose the following prior model: $\mathbf{q}_i \sim \mathcal{B}(\mathbf{\Lambda}_p, \mathbf{V}_p)$, $\mathbf{t}_i \sim \mathcal{N}(\mathbf{0}, \sigma_p^2 \mathbf{I})$ where $\mathbf{\Lambda}_p$, $\mathbf{V}_p$, and $\sigma_p^2$ are the prior model parameters, which are assumed to be known. We then choose the following model for the observed variables:

$$\mathbf{q}_{ij}|\mathbf{q}_i, \mathbf{q}_j \sim \mathcal{B}(\mathbf{\Lambda}, \mathbf{V}(\mathbf{q}_j\bar{\mathbf{q}}_i)), \qquad \mathbf{t}_{ij}|\mathbf{q}_i, \mathbf{q}_j, \mathbf{t}_i, \mathbf{t}_j \sim \mathcal{N}(\boldsymbol{\mu}_{ij}, \sigma^2\mathbf{I}), \tag{5}$$

where $\mathbf{\Lambda}$ is a fixed, $\mathbf{V}$ is a matrix-valued function that will be defined in the sequel; $\boldsymbol{\mu}_{ij}$ denotes the expected value of $\mathbf{t}_{ij}$ provided that the values of the relevant latent variables $\mathbf{q}_i \, \mathbf{q}_j$, $\mathbf{t}_i$, $\mathbf{t}_j$ are known, and has the form: $\boldsymbol{\mu}_{ij} \triangleq \mathbf{t}_j - (\mathbf{q}_j\bar{\mathbf{q}}_i)\mathbf{t}_i(\mathbf{q}_i\bar{\mathbf{q}}_j)$. With this modeling strategy, we are expecting that $\mathbf{t}_{ij}$ would be close to the true translation $\boldsymbol{\mu}_{ij}$ that is a deterministic function of the absolute poses. Our strategy also lets $\mathbf{t}_{ij}$ differ from $\boldsymbol{\mu}_{ij}$ and the level of this flexibility is determined by $\sigma^2$.

Constructing Bingham distribution on any given mode $\mathbf{q} \in \mathbb{S}^3$ requires finding a frame bundle $\mathbb{S}^3 \to \mathcal{F}\mathbb{S}^3$ composed of the unit vector (the mode) and its orthonormals. Being *parallelizable* ($d = 1, 2, 4$ or $8$), manifold of unit quaternions enjoys an injective homomorphism to the orthonormal matrix ring composed of the orthonormal basis [35]. Thus, we define $\mathbf{V} : \mathbb{S}^3 \mapsto \mathbb{R}^{4 \times 4}$ as follows:

$$\mathbf{V}(\mathbf{q}) \triangleq \begin{bmatrix} q_1 & -q_2 & -q_3 & q_4 \\ q_2 & q_1 & q_4 & q_3 \\ q_3 & -q_4 & q_1 & -q_2 \\ q_4 & q_3 & -q_2 & -q_1 \end{bmatrix}.$$ It is easy to verify that $\mathbf{V}(\mathbf{q})$ is orthonormal for every $\mathbf{q} \in \mathbb{S}^3$. $\mathbf{V}(\mathbf{q})$ further gives a convenient notation for representing quaternions as matrices paving the way to linear operations, such as quaternion multiplication or orthonormalization with-

out pesky Gram-Schmidt processes. By using the definition of $\mathbf{V}(\mathbf{q})$ and assuming that the diagonal entries of $\mathbf{\Lambda}$ are sorted in decreasing order, we have the following property:

$$\underset{\mathbf{q}_{ij}}{\arg\max} \left\{ p(\mathbf{q}_{ij}|\mathbf{q}_i, \mathbf{q}_j) = \mathcal{B}(\mathbf{\Lambda}, \mathbf{V}(\mathbf{q}_j\bar{\mathbf{q}}_i)) \right\} = \mathbf{q}_j\bar{\mathbf{q}}_i. \tag{6}$$

Similar to the proposed observation model for the relative translations, given the true poses $\mathbf{q}_i, \mathbf{q}_j$, this modeling strategy sets the most likely value of the relative pose to the deterministic value $\mathbf{q}_j\bar{\mathbf{q}}_i$, and also lets $\mathbf{q}_{ij}$ differ from this value up to the extent determined by $\mathbf{\Lambda}$. This configuration is illustrated in Fig 1(d).

Representing $SE(3)$ in the form of a quaternion-translation parameterization, we can now formulate the motion-synchronization problem as a probabilistic inference problem. In particular we are interested in the following two quantities:

1. The maximum a-posteriori (MAP) estimate: $(\mathbf{Q}^\star, \mathbf{T}^\star) = \arg\max_{\mathbf{Q},\mathbf{T}} p(\mathbf{Q}, \mathbf{T}|\mathcal{D}) =$

$$\underset{\mathbf{Q},\mathbf{T}}{\arg\max} \left( \sum_{(i,j)\in E} \left\{ \log p(\mathbf{q}_{ij}|\mathbf{Q}, \mathbf{T}) + \log p(\mathbf{t}_{ij}|\mathbf{Q}, \mathbf{T}) \right\} + \sum_i \log p(\mathbf{q}_i) + \sum_i \log p(\mathbf{t}_i) \right), \tag{7}$$

where $\mathcal{D} \equiv \{\mathbf{q}_{ij}, \mathbf{t}_{ij}\}_{(i,j)\in E}$ denotes the observations, $\mathbf{Q} \equiv \{\mathbf{q}_i\}_{i=1}^n$ and $\mathbf{T} \equiv \{\mathbf{t}_i\}_{i=1}^n$.

2. The full posterior distribution: $p(\mathbf{Q}, \mathbf{T}|\mathcal{D}) \propto p(\mathcal{D}|\mathbf{Q}, \mathbf{T}) \times p(\mathbf{Q}) \times p(\mathbf{T})$.

Both of these problems are very challenging and cannot be directly addressed by standard methods such as gradient descent (problem 1) or standard MCMC methods (problem 2). The difficulty in these problems is mainly originated by the fact that the posterior density is non-log-concave (i.e. the negative log-posterior is non-convex) and any algorithm that aims at solving one of these problems should be able to operate in the particular manifold of this problem, that is $(\mathbb{S}^3)^n \times \mathbb{R}^{3n} \subset \mathbb{R}^{7n}$.

## 4 Tempered Geodesic Monte Carlo for Pose Graph Optimization

**Connection between sampling and optimization:** In a recent study [36], Liu et al. proposed the stochastic gradient geodesic Monte Carlo (SG-GMC) as an extension to [26] and provided a practical posterior sampling algorithm for the problems that are defined on manifolds whose geodesic flows are analytically available. Since our augmented quaternions form such a manifold[1], we can use this algorithm for generating (approximate) samples from the posterior distribution, which would address the second problem defined in Section 3.

Recent studies have shown that SG-MCMC techniques [37, 38, 39, 40, 41] are closely related to optimization [42, 43, 44, 45, 28, 29] and they indeed have a strong potential in non-convex problems due to their randomized nature. In particular, it has been recently shown that, a simple variant of SG-MCMC is guaranteed to converge to a point near a local optimum in polynomial time [46, 47] and eventually converge to a point near the global optimum [43], even in non-convex settings. Even though these recent results illustrated the advantages of SG-MCMC in optimization, it is not clear how to develop an SG-MCMC-based optimization algorithm that can operate on manifolds. In this section, we will extend the SG-GMC algorithm in this vein to obtain a *parametric* algorithm, which is able to both sample from the posterior distribution and perform optimization for obtaining the MAP estimates depending on the choice of the practitioner. In other words, the algorithm should be able to address both problems that we defined in Section 3 with theoretical guarantees.

We start by defining a more compact notation that will facilitate the presentation of the algorithm. We define the variable $\mathbf{x} \in \mathcal{X}$, such that $\mathbf{x} \triangleq [\mathbf{q}_1^\top, \ldots, \mathbf{q}_n^\top, \mathbf{t}_1^\top, \ldots, \mathbf{t}_n^\top]^\top$ and $\mathcal{X} \triangleq (\mathbb{S}^3)^n \times \mathbb{R}^{3n}$. The posterior density of interest then has the form $\pi_{\mathcal{H}}(\mathbf{x}) \triangleq p(\mathbf{x}|\mathcal{D}) \propto \exp(-U(\mathbf{x}))$ with respect to the Hausdorff measure, where $U$ is called the *potential* energy has the following form: $U(\mathbf{x}) \triangleq -(\log p(\mathcal{D}|\mathbf{x}) + \log p(\mathbf{x})) = -(\log p(\mathcal{D}|\mathbf{Q}, \mathbf{T}) + \log p(\mathbf{Q}) + \log p(\mathbf{T}))$. We define a *smooth embedding* $\xi : \mathbb{R}^{6n} \mapsto \mathcal{X}$ such that $\xi(\tilde{\mathbf{x}}) = \mathbf{x}$. If we consider the embedded posterior density $\pi_\lambda(\tilde{\mathbf{x}}) \triangleq p(\tilde{\mathbf{x}}|\mathcal{D})$ with respect to the Lebesgue measure, then by the area formula (cf. Theorem 1 in [48]), we have the following key property: $\pi_{\mathcal{H}}(\mathbf{x}) = \pi_\lambda(\tilde{\mathbf{x}})/\sqrt{|\mathbf{G}(\tilde{\mathbf{x}})|}$, where $|\mathbf{G}|$ denotes the determinant of the Riemann metric tensor $[\mathbf{G}(\tilde{\mathbf{x}})]_{i,j} \triangleq \sum_{l=1}^{7n} \frac{\partial x_l}{\partial \tilde{x}_i} \frac{\partial x_l}{\partial \tilde{x}_j}$ for all $i, j \in \{1, \ldots, 6n\}$.

The main idea in our approach is to introduce an *inverse temperature* variable $\beta \in \mathbb{R}_+$ and consider the *tempered* posterior distributions whose density is proportional to $\exp(-\beta U(\mathbf{x}))$. When $\beta = 1$, this density coincides with the original posterior; however, as $\beta$ goes to infinity, the tempered density concentrates near the global minimum of the potential $U$ [49, 50]. This important property implies that, for large enough $\beta$, a random sample that is drawn from the tempered posterior would be close to the global optimum and can therefore be used as a MAP estimate.

**Construction of the algorithm:** We will now construct the proposed algorithm. In particular, we will first extend the continuous-time Markov process proposed in [36] and develop a process whose marginal stationary distribution has a density proportional to $\exp(-\beta U(\mathbf{x}))$ for any given $\beta > 0$. Then we will develop practical algorithms for generating samples from this tempered posterior.

We propose the following stochastic differential equation (SDE) in the Euclidean space by making use of the embedding $\xi$:

$$d\tilde{\mathbf{x}}_t = \mathbf{G}(\tilde{\mathbf{x}}_t)^{-1} \mathbf{p}_t dt$$

$$d\mathbf{p}_t = -\left(\nabla_{\tilde{\mathbf{x}}} U_\lambda(\tilde{\mathbf{x}}_t) + \frac{1}{2}\nabla_{\tilde{\mathbf{x}}} \log|\mathbf{G}| + c\mathbf{p}_t + \frac{1}{2}\nabla_{\tilde{\mathbf{x}}}(\mathbf{p}_t^\top \mathbf{G}^{-1}\mathbf{p}_t)dt + \sqrt{(2c/\beta)\mathbf{M}^\top\mathbf{M}}\, dW_t, \quad (8)$$

where $\nabla_{\tilde{\mathbf{x}}} U_\lambda \triangleq -\nabla_{\tilde{\mathbf{x}}} \log \pi_\lambda$, $\mathbf{G}$ and $\mathbf{M}$ are short-hand notations for $\mathbf{G}(\tilde{\mathbf{x}}_t)$ and $[\mathbf{M}(\tilde{\mathbf{x}}_t)]_{ij} \triangleq \partial x_i/\partial \tilde{x}_j$, respectively, $\mathbf{p}_t \in \mathbb{R}^{6n}$ is called the *momentum* variable, $c > 0$ is called the *friction*, and $W_t$ denotes the standard Brownian motion in $\mathbb{R}^{6n}$.

We will first analyze the invariant measure of the SDE (8).

**Proposition 1.** *Let $\boldsymbol{\varphi}_t = [\tilde{\mathbf{x}}_t, \mathbf{p}_t^\top]^\top \in \mathbb{R}^{12n}$ and $(\boldsymbol{\varphi}_t)_{t \geq 0}$ be a Markov process that is a solution of the SDE (8). Then $(\boldsymbol{\varphi}_t)_{t \geq 0}$ has an invariant measure $\mu_\varphi$, whose density with respect to the Lebesgue measure is proportional to $\exp(-\mathcal{E}_\lambda(\boldsymbol{\varphi}))$, where $\mathcal{E}_\lambda$ is is defined as follows:*

$$\mathcal{E}_\lambda(\boldsymbol{\varphi}) \triangleq \beta U_\lambda(\tilde{\mathbf{x}}) + \frac{\beta}{2}\log|\mathbf{G}(\tilde{\mathbf{x}})| + \frac{\beta}{2}\mathbf{p}^\top \mathbf{G}(\tilde{\mathbf{x}})^{-1}\mathbf{p}. \quad (9)$$

All the proofs are given in the supplementary document. By using the area formula and the definitions of $\mathbf{G}$ and $\mathbf{M}$, one can show that the density of $\mu_\varphi$ can also be written with respect to the Hausdorff measure, as follows: (see Section 3.2 in [26] for details) $\mathcal{E}_{\mathcal{H}}(\mathbf{x}, \mathbf{v}) \triangleq \beta U + \frac{\beta}{2}\mathbf{v}^\top \mathbf{v}$, where $\mathbf{v} = \mathbf{M}(\mathbf{M}^\top\mathbf{M})^{-1}\mathbf{p}$. This result shows that, if we could exactly simulate the SDE (8), then the *marginal* distribution of the sample paths would converge to a measure $\pi_{\mathbf{x}}$ on $\mathcal{X}$ whose density is proportional to $\exp(-\beta U(\mathbf{x}))$. Therefore, for $\beta = 1$ we would be sampling from $\pi_{\mathcal{H}}$ (i.e. we recover SG-GMC), and for large $\beta$, we would be sampling near the global optimum of $U$. An illustration of the behavior of $\beta$ on a toy example is provided in the supplementary material.

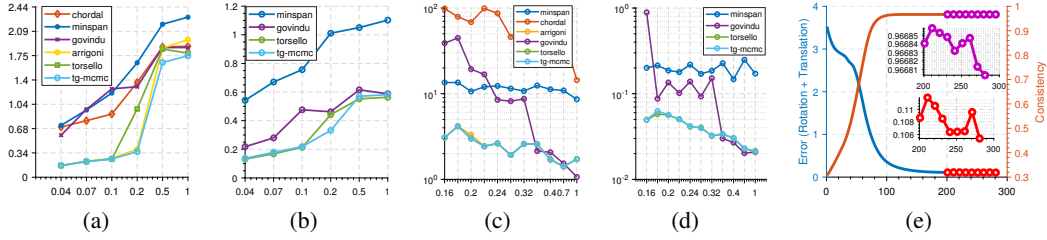

Figure 2: Synthetic Evaluations. **(a)** Mean Riemannian error vs noise variance. **(b)** Mean Euclidean (translational) error vs noise variance. **(c)** Riemannian error vs $e$ for $N = 50$. $e = |E|/N^2$ refers to graph completeness and $N$ to the node count. **(d)** Euclidean error for $N = 50$ vs $e$. **(e)** Monitoring the absolute error w.r.t. ground truth, during optimization and respective posterior sampling.

**Numerical integration:** We will now develop an algorithm for simulating (8) in discrete-time. We follow the approach given in [26, 36], where we split (8) into three disjoint parts and solve those parts analytically in an iterative fashion. The split SDE is given as follows:

$$\text{A:} \begin{cases} d\tilde{\mathbf{x}}_t = \mathbf{G}^{-1}\mathbf{p}_t dt \\ d\mathbf{p}_t = -\frac{1}{2}\nabla(\mathbf{p}_t^\top \mathbf{G}^{-1}\mathbf{p}_t)dt \end{cases} \quad \text{B:} \begin{cases} d\tilde{\mathbf{x}}_t = 0 \\ d\mathbf{p}_t = -c\mathbf{p}_t dt \end{cases} \quad \text{O:} \begin{cases} d\tilde{\mathbf{x}}_t = 0 \\ d\mathbf{p}_t = -(\nabla U_\lambda(\tilde{\mathbf{x}}_t) + \frac{1}{2}\nabla \log|\mathbf{G}|)dt \\ \qquad\qquad + \sqrt{\frac{2c}{\beta}\mathbf{M}^\top \mathbf{M}}dW_t. \end{cases}$$

The nice property of these (stochastic) differential equations is that, each of them can be analytically simulated directly on the manifold $\mathcal{X}$, by using the identity $\mathbf{x} = \xi(\tilde{\mathbf{x}})$ and the definitions of $\mathbf{G}$, $\mathbf{M}$, and $\mathbf{v}$. In practice, one first needs to determine a sequence for the A, B, O steps, set a step-size $h$ for integration along the time-axis $t$, and solve those steps one by one in an iterative fashion [51, 39]. In our applications, we have emprically observed that the sequence BOA provides better results among several other combinations, including the ABOBA scheme that was used in [36]. We provide the solutions of the A, B, O steps, as well as the required gradients in the supplementary material.

**Theoretical analysis:** In this section, we will provide non-asymptotic results for the proposed algorithm. Let us denote the output of the algorithm $\{\mathbf{x}_k\}_{k=1}^N$, where $k$ denotes the iterations and $N$ denotes the number of iterations. In the MAP estimation problem, we are interested in finding $\mathbf{x}^\star \triangleq \arg\min_{\mathbf{x}} U(\mathbf{x})$, whereas for full Bayesian inference, we are interested in approximating posterior expectations with finite sample averages, i.e. $\bar{\phi} \triangleq \int_{\mathcal{X}} \phi(\mathbf{x})\pi_{\mathcal{H}}(\mathbf{x})\, d\mathbf{x} \approx \hat{\phi} \triangleq (1/N)\sum_{k=1}^N \phi(\mathbf{x}_k)$, where $\phi$ is a test function.

As briefly discussed in [36], the convergence behavior of the SG-GMC algorithm can be directly analyzed within the theoretical framework presented in [39]. In a nutshell, the theory in [39] suggests that, with the BOA integration scheme, the bias $|\mathbb{E}\hat{\phi} - \phi|$ is of order $\mathcal{O}(N^{-1/2})$.

In this study, we focus on the MAP estimation problem and analyze the *ergodic* error $\mathbb{E}[\hat{U}_N - U^\star]$, where $\hat{U}_N \triangleq (1/N)\sum_{k=1}^N U(\mathbf{x}_k)$ and $U^\star \triangleq U(\mathbf{x}^\star)$. This error resembles the bias where the test function $\phi$ is chosen as the potential $U$; however, on the contrary, it directly relates the sample average to the global optimum. Similar ergodic error notions have already been considered in non-convex optimization [52, 53, 28]. We present our main result in the following theorem. Due to space limitations and for avoiding obscuring the results, we present the required assumptions and the explicit forms of constants in the supplementary document.

**Theorem 1.** *Assume that the conditions given in the supp. doc. hold. If the iterates are obtained by using the* BOA *the scheme, then the following bound holds for $\beta$ small enough and $\mathcal{X} = (\mathbb{S}^3)^n \times \mathbb{R}^{3n}$:*

$$\left|\mathbb{E}\hat{U}_N - U^\star\right| = \mathcal{O}\big(\beta/(Nh) + h/\beta + 1/\beta\big), \tag{10}$$

*Sketch of the proof.* We decompose the error into two terms: $\mathbb{E}[\hat{U}_N - U^\star] = \mathcal{A}_1 + \mathcal{A}_2$, where $\mathcal{A}_1 \triangleq \mathbb{E}[\hat{U}_N - \bar{U}_\beta]$ and $\mathcal{A}_2 \triangleq [\bar{U}_\beta - U^\star] \geq 0$, and $\bar{U}_\beta \triangleq \int_{\mathcal{X}} U(\mathbf{x})\pi_{\mathbf{x}}(d\mathbf{x})$. The term $\mathcal{A}_1$ is the bias term, which we can bounded by using existing results. The rest of the proof deals with bounding $\mathcal{A}_2$, where we incorporate ideas from [43]. The full proof resides in the supplementary. □

Table 1: Evaluations on EPFL Benchmark.

| | Ozyesil et. al. | | R-GODEC | | Govindu | | Torsello | | EIG-SE(3) | | **TG-MCMC** | |
|---|---|---|---|---|---|---|---|---|---|---|---|---|
| | MRE | MTE | MRE | MTE | MRE | MTE | MRE | MTE | MRE | MTE | MRE | MTE |
| HerzJesus-P8 | 0.060 | 0.007 | 0.040 | 0.009 | 0.106 | 0.015 | 0.106 | 0.015 | 0.040 | 0.004 | 0.106 | 0.015 |
| HerzJesus-P25 | 0.140 | 0.065 | 0.130 | 0.038 | 0.081 | 0.020 | 0.081 | 0.020 | 0.070 | 0.010 | 0.081 | 0.020 |
| Fountain-P11 | 0.030 | 0.004 | 0.030 | 0.006 | 0.071 | 0.004 | 0.071 | 0.004 | 0.030 | 0.004 | 0.071 | 0.004 |
| Entry-P10 | 0.560 | 0.203 | 0.440 | 0.433 | 0.101 | 0.035 | 0.101 | 0.035 | 0.040 | 0.009 | 0.090 | 0.035 |
| Castle-P19 | 3.690 | 1.769 | 1.570 | 1.493 | 0.393 | 0.147 | 0.393 | 0.147 | 1.480 | 0.709 | 0.393 | 0.148 |
| Castle-P30 | 1.970 | 1.393 | 0.780 | 1.123 | 0.631 | 0.323 | 0.629 | 0.321 | 0.530 | 0.212 | 0.622 | 0.285 |
| Average | 1.075 | 0.574 | 0.498 | 0.517 | 0.230 | 0.091 | 0.230 | 0.090 | 0.365 | 0.158 | **0.227** | **0.085** |

Theorem 1 shows that the proposed algorithm will eventually provide samples that are close to the global optimizer $\mathbf{x}^\star$ even when $U$ is non-convex. This result is fundamentally different from the guarantees for the existing convex optimization algorithms on manifolds [54, 55], and is mainly due to the stochasticity of the algorithm that is introduced by the Brownian motion. However, despite this nice theoretical property, in practice our algorithm will still be affected by the *meta-stability phenomenon*, where it will converge near a local minimum and stay there for an exponential amount of time [47].

We also note that our proof covers only the case where $\mathcal{X} = (\mathbb{S}^3)^n \times \mathbb{R}^{3n}$; however, we believe that it can be easily extended to more general setting. We also note that our gradient computations can be replaced with stochastic gradients in the case of large-scale applications where the number of data points can be prohibitively large, so that computing the gradients at each iteration becomes practically infeasible. The same theoretical results hold as long as the stochastic gradients are unbiased.

## 5 Experiments

In a sequel of evaluations, we will be benchmarking our TG-MCMC against the state of the art methods including subsets of: convex programming of Ozyesil et. al. [56], Lie algebraic method of Govindu [15], dual quaternions linearization of Torsello et. al. [15], direct EIG-SE3 method of Arrigoni [12] and R-GODEC [57]. We also include two baseline methods: 1. propagating the pose information along one possible minimum spanning tree, 2. the chordal averaging [58].

**Synthetic Evaluations:** We first synthesize random problems by drawing quaternions from Bingham and translations from Gaussian distributions, and randomly dropping $(100|E|/N^2)\%$ edges from a fully connected pose graph. On these problems, we run a series of tests including monitoring the gradient steps, noise robustness, tolerance to graph completeness (sparsity) and fidelity w.r.t. ground truth. For each test, we distort the graph for the entity we test, i.e. add noise on nodes if we test the noise resilience. The rotational errors are evaluated by the true Riemannian distance, $\|\log(\mathbf{R}^T\hat{\mathbf{R}})\|$, the translations by Euclidean [59]. Fig. 2 plots our findings. It is noticeable that our accuracy is always on par with or better than the state of the art for moderate problems. In presence of increased noise (Figures 2(a), 2(b)) or sparsified graph structure leading to missing data (Figures 2(c), 2(d)), our method shows clear advantage in both rotational and transnational components of the error. This is thanks to our probabilistic formulation and theoretically grounded inference scheme.

**Results in Real Data:** We now evaluate our framework by running SFM on the EPFL Benchmark [60], that provide 8 to 30 images per dataset, along with ground-truth camera transformations. Similar to [12], we use the ground truth scale to circumvent its ambiguity. The mean rotation and translation errors (MRE, MTE) are depicted in Tab. 1. Notice that when rotations and translations are combined, our optimization results in superior minimum for both, not to mention the uncertainty information computed as a by-product. While many methods can perform similarly on easy sets, a clear advantage is visible on Castle sequences where severe noise and missing connections are present. There, for instance, EIG-SE(3) also fails to find a good closed form solution.

Next, we qualitatively demonstrate the unique capability of our method, uncertainty estimation on various SFM problems and datasets [61, 62, 60]. To do so, we first run our optimizer setting $\beta$ to infinity[2] for $> 400$ iterations. After that point, depending on the dataset, we set $\beta$ to a smaller value

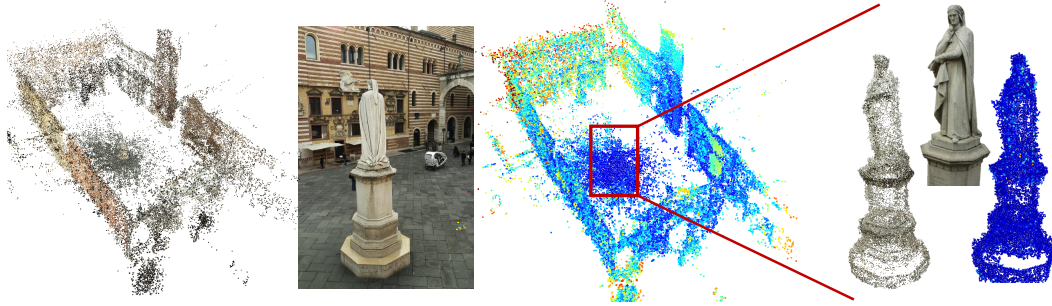

Figure 3: Uncertainty estimation in the Dante Square. From left to right: the colored reconstruction (bundle adjustment used in 3D structure only), a sample image from the dataset, reconstructed points colored w.r.t. uncertainty value, a close-up to the center of the square, Dante statue.

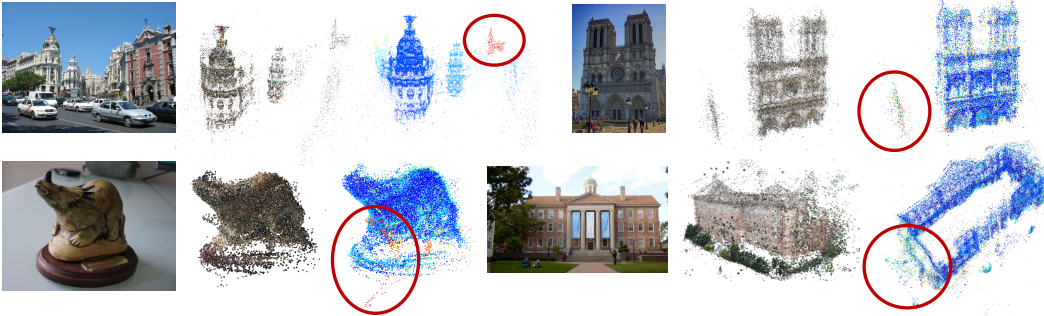

Figure 4: Visualization of uncertainty in Notre Dame, Angel, Dinasour and Fountain datasets.

($\sim 1000$), allowing the sampling of posterior for $40$ times. This behaviour is shown in Fig. 2(e). For each sample, that is a solution of the problem in Eq. 1, we perform a 3D reconstruction, similar to [16]: We first estimate 2D keypoints and relative rotations by running 1) VSFM [63] 2) two-frame bundle adjustment [64, 65] (BA) on image pairs, resulting in pairwise poses, as well as a rough two-view 3D structure. We run our method on these relative poses, computing the absolute estimates. Fixing the estimated poses, a second BA then optimizes for the optimal 3D structure. At the end, we obtain 40 3D scenes per dataset. For each point of each scene, we record the mean and variance across different reconstructions, transferring the uncertainty estimation to the 3D cloud of points. In Figures 3 and 4, we colorize each point by mapping the uncertainty value to RGB space using a jet-colormap, with a scale proportional to the diameter of reconstruction. It is consistently visible that our uncertainty estimates could capture regions of space where there are more and reliable data: Outlying points, noise or distant structures can be identified by interpreting the uncertainty.

## 6   Conclusion

We have proposed TG-MCMC, a manifold-aware, tempered rigid motion synchronization algorithm with a novel probabilistic formulation. TG-MCMC enjoys unique properties of trading-off approximately globally optimal solutions with non-asymptotic guarantees, to drawing samples from the posterior distribution, providing uncertainty estimates for the PGO-initialization problem.

Our algorithm paves the way to a diverse potential future research: First, stochastic gradients can be employed to handle large problems, scaling up to hundreds of thousands of nodes. Next, the uncertainty estimates can be plugged into existing pipelines such as BA or PGO to further improve their quality. We also leave it as a future work to investigate different simulation schemes by altering the order of and combining differently the A, B, and O steps. Finally, TG-MCMC can be extended to different problems, still maintaining its nice theoretical properties.

## Acknowledgements

We would like to thank Robert M. Gower and François Portier for fruitful discussions and Hans Peschke for his feedback and efforts in verifying the correctness of our descriptions. We thank Antonio Vargas of the Mathematics-StackExchange for providing the reference on inequalities for generalized hypergeometric functions. This work is partly supported by the French National Research Agency (ANR) as a part of the FBIMATRIX project (ANR-16-CE23-0014) and by the industrial chair Machine Learning for Big Data from Télécom ParisTech.

## Footnotes

[1]The manifold $(\mathbb{S}^3)^n \times \mathbb{R}^{3n}$ can be expressed as a product of the manifolds $\mathbb{S}^3$ ($n$ times) and $\mathbb{R}^{3n}$. Therefore, its geodesic flow is the combination of the geodesic flows of individual manifolds. Since the geodesic flows in $\mathbb{S}^{d-1}$ and $\mathbb{R}^d$ are analytically available, so is the flow of the product manifold [26].

[2]Note that the case $\beta \to \infty$ renders the SDE degenerate and hence, cannot be analyzed by using our tools. However, due to meta-stability, the algorithm performs similarly either for large $\beta$ or for $\beta \to \infty$.

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
