[Supplementary Material]

# Bayesian Pose Graph Optimization via Bingham Distributions and Tempered Geodesic MCMC
## SUPPLEMENTARY DOCUMENT

**Tolga Birdal**[1,2]     **Umut Şimşekli**[3]     **M. Onur Eken**[1,2]     **Slobodan Ilic**[1,2]

[1] CAMP Chair, Technische Universität München, 85748, München, Germany
[2] Siemens AG, 81739, München, Germany
[3] LTCI, Télécom ParisTech, Université Paris-Saclay, 75013, Paris, France

## 1 Comparison to exponential coordinates

Many optimization algorithms tend to favor Rodrigues vector (exponential coordinates) [1] parameterization due to $R^3$ embedding and geodesics being straight lines [2]. This also leads to simpler Jacobian forms. In this paper, we argue that unit quaternions are more suitable for the approach we pursue: First, 3-vector formulations suffer from infinitely many singularities when the rotation angle approaches 0, $\|\mathbf{a}\| \to 0$, whereas quaternions avoid them [3]. Moreover, quaternions have single redundancy in the representation $\mathbf{q} = -\mathbf{q}$, whereas the normed vectors possess infinite redundancy, i.e. the norm can grow indefinitely, but the angle lies in range $[0 - 2\pi]$. These make it harder to define continuous distributions directly on Rodrigues vectors. Yet, for quaternions there exists the natural antipodally symmetric Bingham distributions.

## 2 Illustration of tempered posteriors

In this study we consider the *tempered* posterior distributions whose density is controlled by the inverse temperature variable $\beta$. When $\beta = 1$, the posterior density coincides with the original posterior; however, as $\beta$ goes to infinity, the tempered density concentrates near the global minimum of the potential $U$ [4, 5]. As we mentioned in the main document, this important property implies that, for large enough $\beta$, a random sample that is drawn from the tempered posterior would be close to the global optimum and can therefore be used as a MAP estimate.

The figure below illustrates this phenomenon on a simple 2-component Gaussian mixture: when $\beta = 1$ we can observe that both modes are visible, but when $\beta = 20$ the mode on the right vanishes and the distribution concentrates around the global mode.

Figure S1: Illustration of tempered posteriors on a simple Gaussian mixture model.

# 3 Numerical Integration

In this section, we provide the details of the numerical integration scheme that was explained in Section 4 of the main document. In short, the overall scheme is an extension of [6], where we introduce the inverse temperature $\beta$.

Once the gradients with respect to the latent variables are computed, i.e.:

$$\nabla_{\mathbf{x}} U(\mathbf{x}) \equiv \{\nabla_{\mathbf{q}_1} U(\mathbf{x}), \dots \nabla_{\mathbf{q}_n} U(\mathbf{x}), \nabla_{\mathbf{t}_1} U(\mathbf{x}), \dots \nabla_{\mathbf{t}_n} U(\mathbf{x})\}, \tag{S1}$$

we can update each of the variables $\mathbf{q}_1, \dots, \mathbf{q}_n, \mathbf{t}_1, \dots, \mathbf{t}_n$ independently from each other, meaning that, the split integration steps, A, B, O can be applied to each of these variables independently. The operations A, B, O will differ depending on the manifold of the particular variable, therefore we will define these operations both on $\mathbb{S}^3$ and $\mathbb{R}^3$ for the two sets of variables $\{\mathbf{q}_i\}_{i=1}^n$ and $\{\mathbf{t}_i\}_{i=1}^n$, respectively.

As we split $\mathbf{x}$ into $\{\mathbf{q}_i\}_{i=1}^n$ and $\{\mathbf{t}_i\}_{i=1}^n$, we similarly split the variable $\mathbf{v}$ into $\{\mathbf{v}_i^{\mathbf{q}}\}_{i=1}^n$ and $\{\mathbf{v}_i^{\mathbf{t}}\}_{i=1}^n$ in order to facilitate the presentation.

## 3.1 Update equations for the rotation components

Set a step-size $h$. For each $\{\mathbf{q}_i, \mathbf{v}_i^{\mathbf{q}}\}$ pairs, the operations A, B, O have the following analytical form:

**Step** A:

Set $\alpha = \|\mathbf{v}_i^{\mathbf{q}}\|$, $\mathbf{q}' \leftarrow \mathbf{q}_i$ and $\mathbf{v}' \leftarrow \mathbf{v}_i^{\mathbf{q}}$.

$$\mathbf{q}_i \leftarrow \mathbf{q}' \cos(\alpha h) + (\mathbf{v}'/\alpha) \sin(\alpha h) \tag{S2}$$

$$\mathbf{v}_i^{\mathbf{q}} \leftarrow -\alpha \mathbf{q}' \sin(\alpha h) + \mathbf{v}' \cos(\alpha h) \tag{S3}$$

**Step** B:

$$\mathbf{v}_i^{\mathbf{q}} \leftarrow \exp(-ch) \mathbf{v}_i^{\mathbf{q}} \tag{S4}$$

**Step** O:

Set $\mathbf{v}' \leftarrow \mathbf{v}_i^{\mathbf{q}}$ and $\mathbf{g} \leftarrow \nabla_{\mathbf{q}_i} U(\mathbf{x})$

$$\mathbf{v}_i^{\mathbf{q}} \leftarrow \mathbf{v}' + P(\mathbf{q}_i)\left(-h\mathbf{g} + \sqrt{2c/\beta}\mathbf{z}_i^{\mathbf{q}}\right), \tag{S5}$$

where $P(\mathbf{q}) = (\mathbf{I} - \mathbf{q}\mathbf{q}^\top)$ denotes the projector and $\mathbf{z}_i^{\mathbf{q}}$ denotes a standard Gaussian random variable on $\mathbb{R}^4$.

## 3.2 Update equations for the translation components

Set a step-size $h$. For each $\{\mathbf{t}_i, \mathbf{v}_i^{\mathbf{t}}\}$ pairs, the operations A, B, O have the following analytical form:

**Step** A:

$$\mathbf{t}_i \leftarrow \mathbf{t}_i + h\mathbf{v}_i^{\mathbf{t}} \tag{S6}$$

**Step** B:

$$\mathbf{v}_i^{\mathbf{t}} \leftarrow \exp(-ch) \mathbf{v}_i^{\mathbf{t}} \tag{S7}$$

**Step** O:

Set $\mathbf{v}' \leftarrow \mathbf{v}_i^{\mathbf{t}}$ and $\mathbf{g} \leftarrow \nabla_{\mathbf{t}_i} U(\mathbf{x})$

$$\mathbf{v}_i^{\mathbf{t}} \leftarrow \mathbf{v}' + \left(-h\mathbf{g} + \sqrt{2c/\beta}\mathbf{z}_i^{\mathbf{t}}\right), \tag{S8}$$

where $\mathbf{z}_i^{\mathbf{t}}$ denotes a standard Gaussian random variable on $\mathbb{R}^3$.

## 3.3 Algorithm pseudocode

We illustrate the overall algorithm in Algorithm 1

---
**Algorithm 1:** TG-MCMC
---
**1 input:** $\mathbf{x}_0 = \{\mathbf{q}_1, \dots, \mathbf{q}_n, \mathbf{t}_1, \dots, \mathbf{t}_n\}$, $\mathbf{v} = \{\mathbf{v}_1^{\mathbf{q}}, \dots, \mathbf{v}_n^{\mathbf{q}}, \mathbf{v}_1^{\mathbf{t}}, \dots, \mathbf{v}_n^{\mathbf{t}}\}$, $\beta$, $c$, $h$
**2 for** $i = 1, \dots, N$ **do**
**3**     Compute the gradient $\nabla_{\mathbf{x}} U(\mathbf{x}_i)$
     // Update the rotation components
**4**     **for** $j = 1, \dots, n$ **do**
**5**       Run the B, O, A steps (in this order) on $\mathbf{q}_j, \mathbf{v}_j^{\mathbf{q}}$ (Section 3.1)
     // Update the translation components
**6**     **for** $j = 1, \dots, n$ **do**
**7**       Run the B, O, A steps (in this order) on $\mathbf{t}_j, \mathbf{v}_j^{\mathbf{t}}$ (Section 3.2)
---

## 4 Assumptions

In this section, we state the assumptions that imply our theoretical results.

**H1.** *The gradient of the potential is Lipschitz continuous, i.e. there exists $L < \infty$, such that $\|\nabla_{\mathbf{x}} U(\mathbf{x}) - \nabla_{\mathbf{x}} U(\mathbf{x}')\| \le L \mathrm{d}_{\mathcal{X}}(\mathbf{x}, \mathbf{x}')$, $\forall \mathbf{x}, \mathbf{x}' \in \mathcal{X}$, where $\mathrm{d}_{\mathcal{X}}$ denotes the geodesic distance on $\mathcal{X}$.*

**H 2.** *The second-order moments of $\pi_{\mathbf{x}}$ are bounded and satisfies the following inequality: $\int_{\mathcal{X}} \|\mathbf{x}\|^2 \pi_{\mathbf{x}}(d\mathbf{x}) \le \frac{C}{\beta}$, for some $C > 0$.*

**H3.** *Let $\psi$ be a functional that is the unique solution of a Poisson equation that is defined as follows:*

$$\mathcal{L}_n \psi(\boldsymbol{\varphi}_n) = U(\mathbf{x}_n) - \bar{U}_{\beta}, \tag{S9}$$

*where $\boldsymbol{\varphi}_n = [\tilde{\mathbf{x}}_n^{\top}, \mathbf{p}_n^{\top}]^{\top}$, $\mathcal{L}_n$ is the generator of (8) at $t = nh$ (see [7] for the definition). The functional $\psi$ and its up to third-order derivatives $\mathcal{D}^k \psi$ are bounded by a function $V(\boldsymbol{\varphi})$, such that $\|\mathcal{D}^k \psi\| \le C_k V^{r_k}$ for $k = 0, 1, 2, 3$ and $C_k, r_k > 0$. Furthermore, $\sup_n \mathbb{E} V^r(\mathbf{x}_n) < \infty$ and $V$ is smooth such that $\sup_{s \in (0,1)} V^r(s\boldsymbol{\varphi} + (1-s)\boldsymbol{\varphi}') \le C(V^r(\boldsymbol{\varphi}) + V^r(\boldsymbol{\varphi}'))$ for all $\boldsymbol{\varphi}, \boldsymbol{\varphi}' \in \mathbb{R}^{12n}$, $r \le \max 2r_k$, and $C > 0$.*

## 5 Proof of Proposition 1

*Proof.* We start by rewriting the SDE given in (8) as follows:

$$d\boldsymbol{\varphi}_t = \left\{ -\left( \underbrace{\begin{bmatrix} 0 & 0 \\ 0 & \frac{c\mathbf{M}^{\top}\mathbf{M}}{\beta}\mathbf{I} \end{bmatrix}}_{\mathbf{D}} + \underbrace{\begin{bmatrix} 0 & -\frac{\mathbf{I}}{\beta} \\ \frac{\mathbf{I}}{\beta} & 0 \end{bmatrix}}_{\mathbf{Q}} \right) \underbrace{\begin{bmatrix} \mathcal{A}(\tilde{\mathbf{x}}_t, \mathbf{p}_t, \beta) \\ \beta\mathbf{G}^{-1}\mathbf{p}_t \end{bmatrix}}_{\nabla_{\boldsymbol{\varphi}}\mathcal{E}_{\lambda}(\boldsymbol{\varphi}_t)} \right\} dt + \sqrt{2\mathbf{D}} dW_t. \tag{S10}$$

where $\mathcal{A}(\tilde{\mathbf{x}}_t, \mathbf{p}_t, \beta) \triangleq \beta\nabla_{\tilde{\mathbf{x}}} U_{\lambda}(\tilde{\mathbf{x}}_t) + \frac{\beta}{2}\nabla_{\tilde{\mathbf{x}}} \log|\mathbf{G}| + \frac{\beta}{2}\nabla_{\tilde{\mathbf{x}}}(\mathbf{p}_t^{\top} \mathbf{G}^{-1}\mathbf{p}_t)$. Here, we observe that $\mathbf{D}$ is positive semi-definite, $\mathbf{Q}$ is anti-symmetric. Then, the desired result is a direct consequence of Theorem 1 of [8]. $\qquad\square$

## 6 Proof of Theorem 1

Before proving Theorem 1, we first prove the following intermediate results, whose proofs are given later in this document.

**Corollary 1.** *Assume that **H1** and **H3** hold. Let $\{\mathbf{x}_n, \mathbf{v}_n\}$ be the output our algorithm with $\beta > 0$. Define $\hat{U}_N \triangleq \frac{1}{N}\sum_{n=1}^{N} U(\mathbf{x}_n)$. Then the following bound holds for the bias:*

$$\left| \mathbb{E}\hat{U}_N - \bar{U}_{\beta} \right| = \mathcal{O}(\frac{\beta}{Nh} + \frac{h}{\beta}). \tag{S11}$$

**Lemma S1.** *Assume that the conditions **H**1 and **H**2 hold. Then, the following bound holds for* $\beta \leq \frac{6}{L\pi^2} \log \frac{CL\pi^4 e}{3n}$:

$$\bar{U}_\beta - U^\star = \mathcal{O}\Big(\frac{1}{\beta}\Big), \tag{S12}$$

*where $C$ is defined in **H**2.*

### 6.1 Proof of Theorem 1

*Proof.* The proof is a direct application of Corollary 1 and Lemma S1. □

## 7 Proof of Corollary 1

*Proof.* From [7][Theorem 2], the bias of a standard SG-MCMC algorithm (i.e. $\beta = 1$) is bounded by

$$\mathcal{O}\big(\frac{1}{Nh'} + \frac{\sum_{n=1}^{N} \|\mathbb{E}\Delta V_n\|}{N} + h'\big). \tag{S13}$$

where $h'$ denotes the step-size and $\Delta V_n$ is an operator and it is related to bias of the stochastic gradient computations if there is any. If the iterates are obtained via full gradient computations $\nabla U$ or unbiased stochastic gradients computations (i.e. the case we consider here), then we have $\|\mathbb{E}\Delta V_n\| = 0$. Then by using a time-scaling argument similar to [9, 10], we define $h = \frac{h'}{\beta}$. This corresponds to running a standard SG-MCMC algorithm directly on the energy function $\mathcal{E}_{\mathcal{H}}(\mathbf{x}, \mathbf{v})$. The result is then obtained by replacing $h'$ by $\frac{h}{\beta}$ in (S13). □

## 8 Proof of Lemma S1

In order to prove Lemma S1, we first need some rather elementary technical results, which we provide in Section 9 for clarity.

*Proof.* We use a similar proof technique to the one given in [9][Proposition 11]. We assume that $\pi_\mathbf{x}$ admits a density, denoted as $\rho(\mathbf{x}) \triangleq \frac{1}{Z_\beta} \exp(-\beta U(\mathbf{x}))$, where $Z_\beta$ is the normalization constant:

$$Z_\beta \triangleq \int_{\mathcal{X}} \exp(-\beta U(\mathbf{x})) d\mathbf{x}. \tag{S14}$$

We start by using the definition of $\bar{U}_\beta$, as follows:

$$\bar{U}_\beta = \int_{\mathcal{X}} U(\mathbf{x})\pi_\mathbf{x}(d\mathbf{x}) = \frac{1}{\beta}(H(\rho) - \log Z_\beta), \tag{S15}$$

where $H(\rho)$ is the *differential entropy*, defined as follows:

$$H(\rho) \triangleq -\int_{\mathcal{X}} \rho(\mathbf{x}) \log \rho(\mathbf{x}) d\mathbf{x}. \tag{S16}$$

We now aim at upper-bounding $H(\rho)$ and lower-bounding $\log Z_\beta$.

By Assumption **H**2, the distribution $\pi_\mathbf{x}$ has a finite second-order moment, therefore all the marginal distributions will also have bounded second order moments. By abusing the notation and denoting $\mathbf{x} \equiv \{\mathbf{q}_1, \ldots, \mathbf{q}_n, \mathbf{t}_1, \ldots, \mathbf{t}_n\}$, and by using the fact that the joint differential entropy is smaller than the sum of the differential entropies of the individual random variables, we can upper-bound $H(\rho)$ as follows:

$$H(\rho) \leq \sum_{i=1}^{n} H(\rho_{\mathbf{q}_i}) + H(\rho_{\mathbf{t}_1,\ldots,\mathbf{t}_n}), \tag{S17}$$

where $\rho_{\mathbf{q}_i}$ denotes the marginal density of $\mathbf{q}_i$ and $\rho_{\mathbf{t}_1,\ldots,\mathbf{t}_n}$ denotes the joint marginal density of $(\mathbf{t}_1, \ldots, \mathbf{t}_n)$. Since $\rho_{\mathbf{t}_1,\ldots,\mathbf{t}_n}$ is defined on $\mathbb{R}^{3n}$, we know that $H(\rho_{\mathbf{t}_1,\ldots,\mathbf{t}_n})$ is upper-bounded by the

differential entropy of a Gaussian distribution on $\mathbb{R}^{3n}$ that has the same second order moment. By denoting the covariance matrix of the Gaussian distribution with $\Sigma$, we obtain:

$$H(\rho_{\mathbf{t}_1,\ldots,\mathbf{t}_n}) \leq \frac{1}{2}\log[(2\pi e)^{3n}\det(\Sigma)] \tag{S18}$$

$$\leq \frac{1}{2}\log[(2\pi e)^{3n}\Big(\frac{\operatorname{tr}(\Sigma)}{3n}\Big)^{3n}] \tag{S19}$$

$$\leq \frac{3n}{2}\log\Big(2\pi e\frac{C}{3\beta n}\Big), \tag{S20}$$

The equations (S19) and (S20) follows by the relation between the arithmetic and geometric means, and Assumption **H**2.

By using a similar argument, since $\rho_{\mathbf{q}_i}$ lives on the unit sphere, its differential entropy is upper-bounded by the differential entropy of the uniform distribution on the unit sphere. Accordingly, we obtain:

$$H(\rho_{\mathbf{q}_i}) \leq \log(2\pi^2), \tag{S21}$$

By using (S20) and (S21) in (S17), we obtain

$$H(\rho) \leq n\log(2\pi^2) + \frac{3n}{2}\log\Big(2\pi e\frac{C}{3\beta n}\Big) \tag{S22}$$

$$= \frac{3n}{2}\log(\sqrt{2}\pi)^{4/3} + \frac{3n}{2}\log\Big(2\pi e\frac{C}{3\beta n}\Big) \tag{S23}$$

$$\leq \frac{3n}{2}\log\Big(\frac{4\pi^3 eC}{3\beta n}\Big). \tag{S24}$$

We now lower-bound $\log Z_\beta$. By definition, we have

$$\log Z_\beta = \log \int_{\mathcal{X}} \exp(-\beta U(\mathbf{x}))d\mathbf{x} \tag{S25}$$

$$= -\beta U^\star + \log \int_{\mathcal{X}} \exp(\beta(U^\star - U(\mathbf{x})))d\mathbf{x} \tag{S26}$$

$$\geq -\beta U^\star + \log \int_{\mathcal{X}} \exp(-\frac{\beta L\pi^2\|\mathbf{x} - \mathbf{x}^\star\|^2}{8})d\mathbf{x} \tag{S27}$$

Here, in (S27) we used Assumption **H**1 and Corollary 2 (presented below). By using $\mathbf{x} \equiv [\mathbf{q}_1^\top,\ldots,\mathbf{q}_n^\top,\mathbf{t}^\top]^\top$ and $\mathbf{x}^\star \equiv [(\mathbf{q}_1^\star)^\top,\ldots,(\mathbf{q}_n^\star)^\top,(\mathbf{t}^\star)^\top]^\top$, and $\mathbf{t} \equiv [\mathbf{t}_1^\top,\ldots,\mathbf{t}_n^\top]^\top$, $\mathbf{t}^\star \equiv [(\mathbf{t}_1^\star)^\top,\ldots,(\mathbf{t}_n^\star)^\top]^\top$ we obtain:

$$\log Z_\beta \geq -\beta U^\star + \log\left(\prod_{i=1}^{n}\int_{\mathbb{S}^3}\exp(-\frac{\beta L\pi^2\|\mathbf{q}_i - \mathbf{q}_i^\star\|^2}{8})d\mathbf{q}_i\right)$$

$$+ \log\left(\int_{\mathbb{R}^{3n}}\exp(-\frac{\beta L\pi^2\|\mathbf{t} - \mathbf{t}^\star\|^2}{8})d\mathbf{t}\right) \tag{S28}$$

$$= -\beta U^\star + \log\left(\prod_{i=1}^{n}\int_{\mathbb{S}^3}\exp(-\frac{\beta L\pi^2\|\mathbf{q}_i - \mathbf{q}_i^\star\|^2}{8})d\mathbf{q}_i\right)$$

$$+ \frac{3n}{2}\log(\frac{4}{\beta L\pi}). \tag{S29}$$

Let us focus on the integral with respect to $\mathbf{q}_i$. By definition, we have:

$$\int_{\mathbb{S}^3}\exp(-\frac{\beta L\pi^2\|\mathbf{q}_i - \mathbf{q}_i^\star\|^2}{8})d\mathbf{q}_i = \int_{\mathbb{S}^3}\exp\Big(-\frac{\beta L\pi^2}{8}(2 - 2\mathbf{q}_i^\top\mathbf{q}_i^\star)\Big)d\mathbf{q}_i \tag{S30}$$

$$= \exp\Big(-\frac{\beta L\pi^2}{4}\Big)\int_{\mathbb{S}^3}\exp\Big(\frac{\beta L\pi^2}{4}\mathbf{q}_i^\top\mathbf{q}_i^\star\Big)d\mathbf{q}_i. \tag{S31}$$

By using the connection between the integral on the right hand side of the above equation and the Von Mises–Fisher distribution [11], we obtain:

$$\int_{\mathbb{S}^3} \exp(-\frac{\beta L \pi^2 \|\mathbf{q}_i - \mathbf{q}_i^\star\|^2}{8}) d\mathbf{q}_i = \exp\left(-\frac{\beta L \pi^2}{4}\right) 2\pi^2 \mathcal{I}_1\left(\frac{\beta L \pi^2}{4}\right) \frac{4}{\beta L \pi^2}, \tag{S32}$$

where $\mathcal{I}_1$ denotes the modified Bessel function of the first kind. By [12] (see Equation 6.25 in the reference), we know that $\mathcal{I}_1(x) \geq x/2$. By using this inequality in (S32), we obtain:

$$\int_{\mathbb{S}^3} \exp(-\frac{\beta L \pi^2 \|\mathbf{q}_i - \mathbf{q}_i^\star\|^2}{8}) d\mathbf{q}_i \geq \exp\left(-\frac{\beta L \pi^2}{4}\right) \pi^2. \tag{S33}$$

We can insert (S33) in (S29), as follows:

$$\log Z_\beta \geq -\beta U^\star + \frac{3n}{2} \log(\frac{4}{\beta L \pi}) + \sum_{i=1}^n \log \int_{\mathbb{S}^3} \exp(-\frac{\beta L \pi^2 \|\mathbf{q}_i - \mathbf{q}_i^\star\|^2}{8}) d\mathbf{q}_i \tag{S34}$$

$$\geq -\beta U^\star + \frac{3n}{2} \log(\frac{4}{\beta L \pi}) - n\frac{\beta L \pi^2}{4} + 2n \log \pi \tag{S35}$$

$$\geq -\beta U^\star + \frac{3n}{2} \log(\frac{4}{\beta L \pi}) - n\frac{\beta L \pi^2}{4} \tag{S36}$$

Finally, by combining (S15), (S24), and (S36), we obtain:

$$\bar{U}_\beta - U^\star = \frac{1}{\beta}(H(\rho) - \log Z_\beta) - U^\star \tag{S37}$$

$$\leq \frac{3n}{2\beta} \log\left(\frac{4\pi^3 eC}{3\beta n}\right) - \frac{3n}{2\beta} \log(\frac{4}{\beta L \pi}) + n\frac{L\pi^2}{4} \tag{S38}$$

$$= \frac{3n}{2\beta} \log\left(\frac{CL\pi^4 e}{3n}\right) + n\frac{L\pi^2}{4} \tag{S39}$$

$$\leq \frac{3n}{\beta} \log\left(\frac{CL\pi^4 e}{3n}\right). \tag{S40}$$

The last line follows from the hypothesis. This finalizes the proof. □

## 9 Technical Results

In the following lemma, we generalize [13][Lemma 1.2.3] to manifolds. Similar arguments can be found in [14, 15].

**Lemma S2.** *Let $\mathcal{X} \subset \mathbb{R}^n$ be a Rimannian manifold with metric $d_{\mathcal{X}}$, and let $\gamma : [0,1] \mapsto \mathcal{X}$ be a constant-speed geodesic curve between two points $\mathbf{x}, \mathbf{y} \in \mathcal{X}$, such that $\gamma(0) = \mathbf{x}$ and $\gamma(1) = \mathbf{y}$. Let $f : \mathcal{X} \mapsto \mathbb{R}$ be a continuously differentiable function with Lipschitz continuous gradients. Then the following bound holds for every $\mathbf{x}, \mathbf{y} \in \mathcal{X}$:*

$$\left| f(\mathbf{y}) - f(\mathbf{x}) - \int_0^1 \langle \nabla f(\mathbf{x}), \gamma'(t) \rangle dt \right| \leq \frac{L}{2} d_{\mathcal{X}}(\mathbf{x}, \mathbf{y})^2, \tag{S41}$$

*where $\langle \mathbf{x}, \mathbf{y} \rangle \triangleq \mathbf{x}^\top \mathbf{y}$ and $L$ denotes the Lipschitz constant.*

*Proof.* Let us define a function $\varphi : [0,1] \mapsto \mathbb{R}$, such that $\varphi(t) \triangleq f(\gamma(t))$. By definition, we have $\varphi(0) = f(\mathbf{x})$ and $\varphi(1) = f(\mathbf{y})$. By using the second fundamental theorem of calculus, we can write:

$$\varphi(1) - \varphi(0) = \int_0^1 \varphi'(t) dt, \tag{S42}$$

where $\varphi'(t)$ denotes the derivative of $\varphi(t)$ with respect to $t$. By the theorem of derivation of composite functions, we have

$$\varphi'(t) = \langle \nabla f(\gamma(t)), \gamma'(t) \rangle. \tag{S43}$$

By combining (S42) and (S43), we obtain the following identity for all $\mathbf{x}, \mathbf{y} \in \mathcal{X}$:

$$f(\mathbf{y}) = f(\mathbf{x}) + \int_0^1 \langle \nabla f(\gamma(t)), \gamma'(t) \rangle dt \tag{S44}$$

$$= f(\mathbf{x}) + \int_0^1 \langle \nabla f(\mathbf{x}), \gamma'(t) \rangle dt + \int_0^1 \langle \nabla f(\gamma(t)) - \nabla f(\mathbf{x}), \gamma'(t) \rangle dt . \tag{S45}$$

Therefore, we obtain

$$\left| f(\mathbf{y}) - f(\mathbf{x}) - \int_0^1 \langle \nabla f(\mathbf{x}), \gamma'(t) \rangle dt \right| = \left| \int_0^1 \langle \nabla f(\gamma(t)) - \nabla f(\mathbf{x}), \gamma'(t) \rangle dt \right| \tag{S46}$$

$$\leq \int_0^1 \left| \langle \nabla f(\gamma(t)) - \nabla f(\mathbf{x}), \gamma'(t) \rangle \right| dt \tag{S47}$$

$$\leq \int_0^1 \| \nabla f(\gamma(t)) - \nabla f(\mathbf{x}) \| \| \gamma'(t) \| dt \tag{S48}$$

$$\leq L \int_0^1 \mathrm{d}_{\mathcal{X}}(\gamma(t), \mathbf{x}) \, \| \gamma'(t) \| dt. \tag{S49}$$

We can now use the fact that the geodesic curve has a constant velocity, such that $\| \gamma'(t) \| = \mathrm{d}_{\mathcal{X}}(\mathbf{x}, \mathbf{y})$ for all $t \in [0,1]$, which also implies $\mathrm{d}_{\mathcal{X}}(\gamma(t_1), \gamma(t_2)) = |t_1 - t_2| \mathrm{d}_{\mathcal{X}}(\gamma(1), \gamma(0))$. Then, using $\mathbf{x} = \gamma(0), \mathbf{y} = \gamma(1)$, we obtain:

$$\left| f(\mathbf{y}) - f(\mathbf{x}) - \int_0^1 \langle \nabla f(\mathbf{x}), \gamma'(t) \rangle dt \right| \leq L \int_0^1 t \mathrm{d}_{\mathcal{X}}(\mathbf{x}, \mathbf{y})^2 dt \tag{S50}$$

$$= \frac{L}{2} \mathrm{d}_{\mathcal{X}}(\mathbf{x}, \mathbf{y})^2. \tag{S51}$$

This concludes the proof. $\qquad\square$

**Corollary 2.** *Under the assumptions of Lemma S2, the following bound holds for all $\mathbf{x} \in \mathcal{X}$*

$$f(\mathbf{x}) - f^\star \leq \frac{L\pi^2}{8} \| \mathbf{x} - \mathbf{x}^\star \|^2, \tag{S52}$$

*where $\mathcal{X} \triangleq (\mathbb{S}^3)^n \times \mathbb{R}^{3n}$, $f^\star = \min_{\mathbf{x}' \in \mathcal{X}} f(\mathbf{x}')$ and $\mathbf{x}^\star = \arg\min_{\mathbf{x}' \in \mathcal{X}} f(\mathbf{x}')$.*

*Proof.* By using Lemma S2 and the obvious facts that $\nabla f(\mathbf{x}^\star) = 0$ and $f(\mathbf{x}) > f^\star$ for all $\mathbf{x} \in \mathcal{X}$, we have:

$$f(\mathbf{x}) - f^\star \leq \frac{L}{2} \mathrm{d}_{\mathcal{X}}(\mathbf{x}, \mathbf{x}^\star)^2. \tag{S53}$$

The inequality given in Equation A.1.1 in [16] states that the geodesic distance on the sphere is bounded by the 2-norm, more precisely, for all $\mathbf{q}, \mathbf{q}' \in \mathbb{S}^{d-1}$ we have:

$$\mathrm{d}_{\mathbb{S}^{d-1}}(\mathbf{q}, \mathbf{q}') \leq \frac{\pi}{2} \| \mathbf{q} - \mathbf{q}' \|. \tag{S54}$$

Using $\mathbf{x} \equiv [\mathbf{q}_1^\top, \dots, \mathbf{q}_n^\top, \mathbf{t}_1^\top, \dots, \mathbf{t}_n^\top]^\top$ and $\mathbf{x}^\star \equiv [(\mathbf{q}_1^\star)^\top, \dots, (\mathbf{q}_n^\star)^\top, (\mathbf{t}_1^\star)^\top, \dots, (\mathbf{t}_n^\star)^\top]^\top$ yields:

$$f(\mathbf{x}) - f^\star \leq \frac{L}{2} \left( \sum_{i=1}^n \mathrm{d}_{\mathbb{S}^3}(\mathbf{q}_i, \mathbf{q}_i^\star)^2 + \sum_{i=1}^n \| \mathbf{t}_i - \mathbf{t}_i^\star \|^2 \right) \tag{S55}$$

$$\leq \frac{L}{2} \left( \frac{\pi^2}{4} \sum_{i=1}^n \| \mathbf{q}_i - \mathbf{q}_i^\star \|^2 + \sum_{i=1}^n \| \mathbf{t}_i - \mathbf{t}_i^\star \|^2 \right) \tag{S56}$$

$$\leq \frac{L\pi^2}{8} \left( \sum_{i=1}^n \| \mathbf{q}_i - \mathbf{q}_i^\star \|^2 + \sum_{i=1}^n \| \mathbf{t}_i - \mathbf{t}_i^\star \|^2 \right) \tag{S57}$$

$$= \frac{L\pi^2}{8} \| \mathbf{x} - \mathbf{x}^\star \|^2. \tag{S58}$$

This concludes the proof. $\qquad\square$

## 10 Gradients of Likelihood and Prior Terms

In this section we provide derivations of the gradients for data and prior terms. For completeness, we find it worthy to once again repeat our MLE formulation:

$$\underset{\mathbf{Q},\mathbf{T}}{\arg\max}\Big(\sum_{(i,j)\in E}\big\{\log p(\mathbf{q}_{ij}|\mathbf{Q},\mathbf{T})+\log p(\mathbf{t}_{ij}|\mathbf{Q},\mathbf{T})\big\}+\sum_i\log p(\mathbf{q}_i)+\sum_i\log p(\mathbf{t}_i)\Big). \quad \text{(S59)}$$

We begin by deriving the gradients of the **rotational components** first, and translations second. In the setting where $\mathbf{V}$ is constant w.r.t. $\mathbf{q}$ the gradient of log Bingham distribution w.r.t. the random variable $\mathbf{q}$ reads:

$$\nabla_{\mathbf{x}}\log\mathcal{B}(\mathbf{x};\boldsymbol{\Lambda},\mathbf{V})=\nabla_{\mathbf{x}}\log\frac{1}{F}+\nabla_{\mathbf{x}}(\mathbf{x}^T\mathbf{V}\boldsymbol{\Lambda}\mathbf{V}^T\mathbf{x})=(\mathbf{V}\boldsymbol{\Lambda}\mathbf{V}^T+\mathbf{V}\boldsymbol{\Lambda}^T\mathbf{V}^T)\mathbf{x}=2\mathbf{V}\boldsymbol{\Lambda}\mathbf{V}^T\mathbf{x}. \quad \text{(S60)}$$

As the first column of $\mathbf{V}$ is the mode, we avoid storing it. Thus, for the implementation purposes, we abuse the notation and use $\mathbf{V}\in\mathbb{R}^{4\times3}$ and $\boldsymbol{\Lambda}\in\mathbb{R}^{3\times3}$. Normalizing constant $F$ drops as it depends on $\boldsymbol{\Lambda}$ only [17]. We also have cases where $\mathbf{V}$ is a function of the mode $\mathbf{q}$, $\mathbf{V}\to\mathbf{V}(\mathbf{q})$. Then:

$$\nabla_{\mathbf{q}}\log\mathcal{B}(\mathbf{x};\boldsymbol{\Lambda},\mathbf{V}(\mathbf{q}))=\nabla_{\mathbf{q}}(\mathbf{x}^T\mathbf{V}(\mathbf{q})\boldsymbol{\Lambda}\mathbf{V}^T(\mathbf{q})\mathbf{x})=\nabla_{\mathbf{k}}(\mathbf{k}^T\boldsymbol{\Lambda}\mathbf{k})\,\nabla_{\mathbf{q}}(\mathbf{k})=2\mathbf{k}^T\boldsymbol{\Lambda}\,\nabla_{\mathbf{q}}(\mathbf{k}), \quad \text{(S61)}$$

where $\mathbf{k}=\mathbf{V}^T(\mathbf{q})\mathbf{x}\in\mathbb{R}^3$ is used to ease the computations. Note that in our particular application it is the case that $\mathbf{x}\leftarrow\mathbf{q}_{ij}$, i.e. the data is specified by the relative poses attached to the edges of the graph. We then speak of the gradient of $\log(p(\mathbf{q}_{ij}\,|\,\mathbf{q}_i,\mathbf{q}_j))$ with $\mathbf{V}\to\mathbf{V}(\mathbf{q}_j\bar{\mathbf{q}}_i)$ w.r.t. $\mathbf{q}_i$. We shorten $\mathbf{r}\leftarrow\mathbf{q}_j\bar{\mathbf{q}}_i$ and write $\mathbf{V}$ as a function of $\mathbf{r}$, $\mathbf{V}(\mathbf{r})\triangleq\mathbf{V}(\mathbf{q}_j\bar{\mathbf{q}}_i)$. Then:

$$\nabla_{\mathbf{q}_i}\log\mathcal{B}(\mathbf{x};\boldsymbol{\Lambda},\mathbf{V}(\mathbf{r}))=\nabla_{\mathbf{r}}\log\mathcal{B}(\mathbf{x};\boldsymbol{\Lambda},\mathbf{V}(\mathbf{r}))\nabla_{\mathbf{q}_i}(\mathbf{r})=2\mathbf{k}^T\boldsymbol{\Lambda}\,\nabla_{\mathbf{r}}(\mathbf{k})\,\nabla_{\mathbf{q}_i}(\mathbf{r}), \quad \text{(S62)}$$

this time with $\mathbf{k}=\mathbf{V}^T(\mathbf{r})\mathbf{x}\in\mathbb{R}^3$. Note that $\nabla_{\mathbf{r}}\log\mathcal{B}(\mathbf{x};\boldsymbol{\Lambda},\mathbf{V}(\mathbf{r}))$ is expanded as in Eq. S61. Using the definition of $\mathbf{V}$ in Eq. 3, the terms simplify to:

$$\mathbf{k}=\begin{bmatrix}q_1x_2-q_2x_1+q_3x_4-q_4x_3\\q_1x_3-q_3x_1-q_2x_4+q_4x_2\\-q_1x_4-q_2x_3+q_3x_2+q_4x_1\end{bmatrix}\qquad\nabla_{\mathbf{q}}(\mathbf{k})=\begin{bmatrix}x_2&-x_1&x_4&-x_3\\x_3&-x_4&-x_1&x_2\\x_4&x_3&-x_2&-x_1\end{bmatrix}. \quad \text{(S63)}$$

The last term in eq. S62 expands as:

$$\nabla_{\mathbf{q}_i}(\mathbf{q}_j\bar{\mathbf{q}}_i)=\begin{bmatrix}q_{j,1}&q_{j,2}&q_{j,3}&q_{j,4}\\q_{j,2}&-q_{j,1}&q_{j,4}&-q_{j,3}\\q_{j,3}&-q_{j,4}&-q_{j,1}&q_{j,2}\\q_{j,4}&q_{j,3}&-q_{j,2}&-q_{j,1}\end{bmatrix}. \quad \text{(S64)}$$

Due to the symmetry of the relative poses in the graph, we do not need to compute the gradients w.r.t. $\mathbf{q}_j$. We will now derive the gradients for **translational components**. Similarly, we start by the gradient of the log likelihood w.r.t. the data. While a shorter derivation through matrix calculus is also possible, we deliberately provide a longer version, as it might be more intuitive:

$$\begin{aligned}\nabla_{\mathbf{t}}\log\mathcal{N}\Big(\mathbf{t};\boldsymbol{\mu},\sigma^2\mathbf{I}\Big)&=\nabla_{\mathbf{t}}\log\frac{1}{G}+\nabla_{\mathbf{t}}\big(-\frac{1}{2}(\mathbf{t}-\boldsymbol{\mu})^T\boldsymbol{\Sigma}^{-1}(\mathbf{t}-\boldsymbol{\mu})\big)\\&=\nabla_{\mathbf{t}}\Big(-\frac{1}{2}\big(\mathbf{t}^T\boldsymbol{\Sigma}^{-1}\mathbf{t}-\mathbf{t}^T\boldsymbol{\Sigma}^{-1}\boldsymbol{\mu}-\boldsymbol{\mu}^T\boldsymbol{\Sigma}^{-1}\mathbf{t}+\boldsymbol{\mu}^T\boldsymbol{\Sigma}^{-1}\boldsymbol{\mu}\big)\Big)\\&=-\frac{1}{2}\big(\mathbf{t}^T(\boldsymbol{\Sigma}^{-1}+\boldsymbol{\Sigma}^{-T})-(\boldsymbol{\Sigma}^{-1}\boldsymbol{\mu})^T-(\boldsymbol{\mu}^T\boldsymbol{\Sigma}^{-1})+0\big)\\&=-\frac{1}{2}\big(2\mathbf{t}^T\boldsymbol{\Sigma}^{-1}-2\boldsymbol{\mu}^T\boldsymbol{\Sigma}^{-1}\big)=(\boldsymbol{\mu}^T-\mathbf{t}^T)\boldsymbol{\Sigma}^{-1}\end{aligned} \quad \text{(S65)}$$

The normalizing constant drops similarly as it does not depend on $\mathbf{t}$.

Similar to rotational counterpart, our algorithm centers the data on the mean of the distribution, also requiring to compute the gradients w.r.t. the mean of the distribution. With a derivation similar to but simpler from Eq. S65, it follows:

$$\nabla_{\boldsymbol{\mu}}\log\mathcal{N}\Big(\mathbf{x};\boldsymbol{\mu},\sigma^2\mathbf{I}\Big)=-\frac{1}{2}\big(2\boldsymbol{\mu}^T\boldsymbol{\Sigma}^{-1}-2\mathbf{x}^T\boldsymbol{\Sigma}^{-1}\big)=(\mathbf{x}^T-\boldsymbol{\mu}^T)\boldsymbol{\Sigma}^{-1} \quad \text{(S66)}$$

When we center the distribution on the data, substituting $\boldsymbol{\mu} \leftarrow \mathbf{t}_j - \mathbf{rt}_i\bar{\mathbf{r}}$, where $\mathbf{r} \leftarrow \mathbf{q}_j\bar{\mathbf{q}}_i$, $\mathbf{x} \leftarrow \mathbf{t}_{ij}$, we arrive at:

$$\nabla_{\mathbf{t}_i} \log \mathcal{N}\left(\mathbf{x}; \boldsymbol{\mu}, \sigma^2 \boldsymbol{I}\right) = \nabla_{\boldsymbol{\mu}} \log \mathcal{N}\left(\mathbf{x}; \boldsymbol{\mu}, \sigma^2 \boldsymbol{I}\right) J_{\mathbf{t}_i}(\boldsymbol{\mu}) \tag{S67}$$

Note that the first term of the right hand side is given in Eq. S66. The second one can be computed from the derivative of the sandwich action on the 1-blade $\mathbf{t}_i$. With slight abuse of notation, in the following we assume that translation-quaternion is purified: $\mathbf{t}_i \leftarrow [0; \mathbf{t}_i]$.

$$J_{\mathbf{t}_i}(\boldsymbol{\mu}) = J_{\mathbf{t}_i}(\mathbf{t}_j - \mathbf{rt}_i\bar{\mathbf{r}}) \tag{S68}$$

$$= -J_{\mathbf{t}_i}\left(Q(\bar{\mathbf{r}})Q(\mathbf{t}_i)\mathbf{r}^T\right) \tag{S69}$$

$$= -J_{\mathbf{t}_i}\left(\left(\mathbf{q}_{ij} \otimes Q(\bar{\mathbf{r}})\right) \text{vec}\left(Q(\mathbf{t}_i)\right)\right) \tag{S70}$$

$$= -J_{\mathbf{t}_i}\left(\mathbf{K} \text{vec}\left(Q(\mathbf{t}_i)\right)\right) \tag{S71}$$

$$= -\mathbf{K}\nabla_{\mathbf{t}_i} \text{vec}\left(Q(\mathbf{t}_i)\right) \tag{S72}$$

$$= -\mathbf{K}\mathbf{J}_{\mathbf{t}_i} \tag{S73}$$

$$= \begin{bmatrix} 0 & 0 & 0 \\ -q_1^2 - q_2^2 + q_3^2 + q_4^2 & 2q_1q_4 - 2q_2q_3 & -2q_1q_3 - 2q_2q_4 \\ -2q_1q_4 - 2q_2q_3 & -q_1^2 + q_2^2 - q_3^2 + q_4^2 & 2q_1q_2 - 2q_3q_4 \\ 2q_1q_3 - 2q_2q_4 & -2q_1q_2 - 2q_3q_4 & -q_1^2 + q_2^2 + q_3^2 - q_4^2 \end{bmatrix} \tag{S74}$$

where $\mathbf{K} \in R^{4 \times 16}$ refers to the Kronecker product matrix $\mathbf{K} = \mathbf{r} \otimes Q(\bar{\mathbf{q}}_{ij})$, $\mathbf{J}_{\mathbf{t}_i} = \nabla_{\mathbf{t}_i} \text{vec}\left(Q(\mathbf{t}_i)\right)$ is the $16 \times 3$ Jacobian matrix and $\text{vec}(\cdot)$ denotes the linearization operator. Individual components $q_i$ belong to $\mathbf{r} = [q_1, q_2, q_3, q_4]$. The map $Q(\cdot) : \mathbb{H}_1 \rightarrow R^{4x4}$ constructs a Quaternion matrix form:

$$Q(\mathbf{q}) = \begin{bmatrix} q_1 & -q_2 & -q_3 & -q_4 \\ q_2 & q_1 & q_4 & -q_3 \\ q_3 & -q_4 & q_1 & q_2 \\ q_4 & q_3 & -q_2 & q_1 \end{bmatrix} \tag{S75}$$

leading to a more compact notation of the quaternion product. In fact this is not very different from the definition of $\mathbf{V}(\mathbf{q})$, as one is free to pick any of the 48 distinct representations out of the matrix ring $\mathbb{M}(4, \mathbb{R})$. For a more thorough reading on differentiating the quaternions, we refer the reader to [18]. Note that Eq. S74 has zeros in the initial row. This is due to the property that all the operations respect the purity of the blade. The final Jacobian matrix can be extracted from the remaining three rows corresponding to the vector part.

Last but not least, we require the gradients of the translational part w.r.t. the quaternions due to the coupling:

$$\nabla_{\mathbf{q}_i} \log \mathcal{N}\left(\mathbf{x}; \boldsymbol{\mu}, \sigma^2 \boldsymbol{I}\right) = \nabla_{\boldsymbol{\mu}} \log \mathcal{N}\left(\mathbf{x}; \boldsymbol{\mu}, \sigma^2 \boldsymbol{I}\right) J_{\mathbf{q}_i}(\boldsymbol{\mu}) \tag{S76}$$

The first term in Eq. S76 is given above and the derivation of the rightmost Jacobian is subject to an operation similar to the ones given in S68.

## 11  Additional Experiments and Details

**Hyper-parameter Selection**  Throughout all the experiments we set $c \leftarrow 1000$ and during optimization $\beta \leftarrow \infty$. In practice, we only set $\beta$ to a very large finite value. $h$ varies between $0.001 - 0.008$ depending on the dataset $((\lambda, \sigma^2))$. We typically set Bingham and Gaussian variances to be at the noise level of the dataset, evaluated empirically. The variance of the Bingham distribution is $\lambda \in [350, 900]$. Likewise, variance of the Gaussian lies in $\sigma^2 \in [0.01, 0.1]$. To show that the choice is not critical, in Fig. S2, we plot $\lambda$, our most sensitive parameter, against the error

Figure S2: Effect of $\lambda$ on rotational ($\mathbf{Q}$) and translational ($\mathbf{T}$) errors.

Figure S3: Robustness to outliers. With respect to the outlier percentage, we plot: **(a)** deviations of rotations from ground truth (mean error) **(b)** deviations of translation from ground truth (mean error) **(c)** graph consistency (for definition, see paper).

Figure S4: Synthetic evaluations against projected gradient descent (PGD). **(a)** Iterations vs negative log likelihood **(b)** Iterations vs absolute rotation error of the estimates w.r.t. ground truth. **(c)** Iterations vs consistency (for definition, see paper).

attained at convergence for different datasets, including synthetic and real. The figure indicates that for a variety of choices, $\lambda > 100$, the solution can safely be found. Note that certain level of noise can also be compensated by the step size, as variance and step size are multiplicative factors. Moreover, the step size can be adjusted dynamically proportional to the dataset size. The number of integration steps varies, typically in range $[350, 800]$ and TG-MCMC runs until convergence.

**Graph Consistency**   In the paper, as well as in this supplementary material, we speak of *graph consistency*, an intuitive measure of quantifying how well the estimated parameters agree to the input data. This measure is easier to interpret than, say, average rotational distance, that is always unit bound. We define the graph consistency as follows:

$$g_c = 1 - \frac{1}{\pi|E|} \sum_{(i,j) \in E} 2 \arccos(\mathbf{q}_{ij}(\mathbf{q}_i \bar{\mathbf{q}}_j)) \tag{S77}$$

In other words, $g_c$ measures how well the relative poses computed from absolute estimates align with the ones given in the data. $g_c = 1$ for the perfect ground truth and $g_c \to 0$ when all estimates are off by $\pi$.

## 11.1   Quantitative Evaluations

**Outliers**   Even though TG-MCMC has no explicit treatment of outliers, it is still of interest to observe the robustness to outliers. We do that synthetically, by following a similar experimentation setup to the main paper. This time, we increase the outlier ratio in the pose graph by excessively corrupting some of the relative transformation matrices by composing it with random rotations in the range $[60°80°]$ around random axes, and random translations between $[0, 1]$. We then run TG-MCMC, as well as the other algorithms under consideration. Our results are depicted in Fig. S3. Many state-of-the-art methods that lack outlier handling are similar in performance. However, advantage of TG-MCMC is more apparent as the outlier percentage increases.

**Projected Gradient Descent**   Next, we compare our method against projected gradient descent (PGD) algorithm, that is heavily used when one avoids the manifold operations of quaternions. This amounts to solving our MAP estimation using a standard first order method and projecting the intermediary solutions back onto the manifold. Using compatible step sizes, Fig. S4 plots multiple quantities as iterates progress. It is clearly visible that walking on the manifold is advantageous both in finding quicker solutions (a,b) and reducing the energy of the cost (c).

**Runtime Performance**   We now provide, in Tab. 1a runtime analysis. It is common to all PGO initialization algorithms to outperform BA in terms of speed, which justifies the attempt to initialize PGO. Among the compared methods, we are not the fastest. Our runtimes are just comparable to those of the state of the art. However, we have, in addition, the component of uncertainty estimates, which cannot be provided by the competing algorithms. It is also clearly visible that BA benefits from the TG-MCMC initialization by an order of magnitude (shown in Gains column).

Table 1: Runtimes (seconds) of different algorithms. BA-X refers to BA with initialization X. *Points, Poses* refer to optimizing only points or only poses, respectively.

| | PGO | | | | BA-MinSpan | | BA-Ours | | |
|---|---|---|---|---|---|---|---|---|---|
| Dataset | Arrigoni | Torsello | Govindu | Ours | Points | Poses | Points | Poses | Gains |
| Madrid | 0.137 | 2.255 | 3.033 | 5.794 | 53.82 | 139.6 | 33.30 | 14.54 | **9.60x** |
| South Building | 0.060 | 0.784 | 1.213 | 1.986 | 46.18 | 108.0 | 6.087 | 4.969 | **21.74x** |

## 11.2   Qualitative Evaluations

**Uncertainty Estimation**   We now give further visualizations showing the behavior of uncertainty estimation. Let us first supplement the main paper, by providing close up and easier to view shots of some of the scenes. Due to space limitations, we had to omit some of the larger drawings from the main paper. Fig. S5 illustrates the uncertainty mapping on the Madrid Metropolis reconstruction and Fig. S6 on the South Building dataset [19, 20]. In the former, distant content, which is intrinsically less accurate to triangulate, appears less certain than the structure nearby. This overlaps well with the findings of stereo vision where baseline-to-distance ratio determines the triangulation accuracy. In the latter, though, we see that hard to match content such as vegetation has more uncertainty. This is also natural, because such image regions render the feature matching difficult. Finally, we provide uncertainty maps for two more objects Angel and Fountain. In these objects, due to the small size and noise, uncertainty variation is less apparent and hard to observe. However, on Angel, our algorithm overall manages to spot the noisy points and mark them with higher uncertainty. On the Fountain, the structure close to the borders of the image are shot from a fewer number of cameras, which is what TG-MCMC has discovered.

**(a)** Madrid Metropolis                **(b)** 3D Reconstruction                **(c)** Uncertainty Map

Figure S5: Reconstruction of Madrid Metropolis. Our uncertainty map can reveal the distant structure such as buildings because the 3D triangulation quality decreases with the distance. Samples produced by TG-MCMC can successfully explain these variations.

**Graph Evolution**   To shade light upon the inner workings of TG-MCMC, we now visualize the evolution of the pose graph as iterations/time proceed(s). Fig. S8 presents snapshots of the pose graph of Angel dataset, evolving towards the solution. Notice, our algorithm can start from a random

South Building        Reconstructions        Uncertainty Map

Figure S6: Reconstruction of South Building of UNC. Notice that hard-to-reconstruct structure such as vegetation is also marked to be uncertain by our algorithm, whereas rigid structures such as building façades enjoy high certainty.

Figure S7: Uncertainty visualizations on Angel and Fountain objects.

T=1       T=31       T=111       T=201       T=501

T=901       T=1101       T=1251       Final Poses

Figure S8: Evolution of the graph structure on the Angel object.

initialization and achieve results that are very close to the ground truth. In fact, we also show comparisons of the obtained pose graph against the ground truth poses in Fig. S9. Our low numbers in quantitative error well transfers to qualitative evaluations.

**Further Visual Results from the Used Datasets**    In order to have a better idea of the nature of the datasets we utilized, it is worthwhile to visualize the camera locations as well as the 3D reconstruction following a full bundle adjustment, that optimizes both 3D points (structure) and 3D poses (motion).

**(i)** Ground truth pose graph   **(ii)** Computed pose graph      **(i)** Ground truth pose graph   **(ii)** Computed pose graph

**(a)** Angel                        **(b)** Dinasour

Figure S9: Comparing the resulting pose graph with the ground truth for Angel and Dino objects.

In Fig. S10, we report 6 such visualizations on 3 outdoor, large scenes, as well as 3 object-scanning scenarios. These plots are not the outcome of our approach, but meant as a reference for the datasets we deal with.

**(a)** Dante          **(b)** Madrid Metropolis          **(c)** Notre Dame

**(d)** Dinasour               **(e)** Angel               **(f)** Temple

Figure S10: Results of the full bundle adjustment (structure + camera poses) on several datasets.