[Reviews · NeurIPS 2018]

Reviewer 1



summary: 7/10 This paper proposes a tempered geodesic MCMC approach for bayesian pose graph optimization (PGO). One tempered sampling algorithm is proposed which bridges the optimization and sampling, with its sampling error bound analyzed. Two set of experiments (synthetic and real) are conducted, showing that the accuracy is comparable or better than baselines. One thing noted is, the motivation of the work is PGO post processing takes are time consuming for online /soft-realtime operations. But this question was not answered in the work, i.e., the computation cost was not addressed for the proposed algorithm. quality: 6/10 One concern is mentioned in summary: the motivation is to enable PGO to be realtime/online but the computation is not analyzed/addressed in the work. When descretizing the continuous SDE to sampling steps, there are a few parameters like step sizes and ordering as mentioned. Are those mostly decided by trial-and-error? That much degrades the practical impact of the proposed algorithm. clarity: 8/10 The work is overall clearly presented. originality: 8/10 A decent geodesic MCMC algorithm is provided, not novel in itself, but first time applied to the problem considered. significance: 7/10 PGO is an important problem itself. And one Bayesian algorithm is designed with some tools integrated/constructed. This work is interesting to a wide group of researchers. Updated: As responded in author feedback, the computation is reasonable compared to alternatives. It is suggest to add the table/exp to the supplement as it is tightly related to the motivation.

Reviewer 2



This paper presents a stochastic gradient Monte Carlo approach defined on a Cartesian product of SE(3), a domain commonly used to characterize problems in structure-from-motion (SFM) among other areas. The algorithm is parameterized by an inverse temperature such that when the value goes to inifinity, the algorithm is implicitly operating on a delta function with it's peak at the maximum of the base distribution. The proposed algorithm is formulated as a SDE and a splitting scheme is proposed to integrate it. A theoretical analysis on the SDE and its discretization is explored, showing that 1) the resulting Markov process has the appropriate invariant distribution and 2) the sampler will draw samples close to the maximum of the posterior (in terms of expectation of the unnormalized log posterior). Along with the algorithm, a model is defined using the Bingham distribution to characterize typical SFM posteriors which is then used to perform experiments with the algorithm. The algorithm is validated on synthetic experiments along with some real SFM datasets. The problem that is considered (sampling and optimization over pose graphs on SE(3)^n) is an important one. The proposed algorithm seems suitable and has at least a small amount of novelty. However, there are some problems with both the theoretical and experimental analysis which are detailed below. The paper has potential, but the issues need to be addressed before it is suitable for publication. - The theorem as stated in the paper has omitted a key condition which somewhat limits its applicability to the scenario indicated in the paper. Specifically, according to Lemma S1, the theorem only applies for values of \beta \leq D for some finite constant D. This is not discussed at all in the main paper and only shows up in the statement of the lemma in the supplementary. Importantly, it undermines a major line of argument for the paper, namely that the method provides a mechanism for either MCMC sampling from the posterior *and* a way to find the maximum of the posterior by setting \beta = \infty. As far as I've understood the results, the proof of the theorem is not valid as \beta approaches \infty. - The paper makes claims which are unsubstantiated by the theoretical results. In particular, lines 248-250 claims that the theorem shows that the algorithm will sample close to the global optimizer x^*. This is not what is actually said by the theorem. Rather the theorem states that expected value of U(x) = -\log(p(x)) will be close to U(x^*). Note that this is significant in the case where the posterior non-convex and may be multimodal which line 248 specifically mentions. - A discussion of the tuning of several key hyperparameters is missing in the paper, limiting reproducibility. In particular, how are N (number of integration steps) and h (step size) set experimentally? Are they tuned per experiment or fixed? How sensitive are the results to these (and other) hyperparameters. Similarly, how exactly is the synthetic data generated? Is source code going to be released? This will address some (though not all) of these concerns. There are also several other less significant issues. - Figure 2 (in particular 2(e)) is not well explained, 2e in particular is never even referenced and the caption doesn't explain it clearly. - Fig 2e implies and Lines 284-5 discuss combining error in rotation with error in translation. How is this done? === UPDATE: I have read and considered the authors feedback and would not argue against acceptance of this paper. However, I remain concerned about the looseness of the arguments around the applicability of the theoretical results and, in particular, the disconnect between how they're presented in the paper vs how they're actually defined in the appendix.

Reviewer 3



The paper proposes a principled approach to motion synchronization by formulating it as a probabilistic inference problem and devising an original MCMC sampler to address it. The methods are presented in good detail, and theoretical justification for choices made by the authors is provided. The probabilistic approach allows for uncertainty estimation as well, which is a strong point of this work. Overall I think this is a good paper, however I found that perhaps the experimental results could have been presented better and commented upon in greater detail. For instance, in Table 1, where evaluation on the EPFL benchmark is done, the results on Castle-P19 show very similar performance for the proposed method, the Govindu method and the Torsello method. It would have been interesting to understand the reasons for such very close performance alignment, given this is not the case for the other three comparison methods. Another comment is on the inclusion of the “Average” row, which does not seem to be necessary as this is a comparison amongst problems of different complexity.